



# Vertical Profiles of Liquid Water Content in fog layers during the SOFOG3D experiment

Théophane Costabloz [1], Frédéric Burnet [1], Christine Lac [1], Pauline Martinet [1], Julien Delanoë [2], Susana Jorquera [2], and Maroua Fathalli [1]

[1]CNRM, Université de Toulouse,Météo-France, CNRS, Toulouse, France
[2]LATMOS/IPSL, UVSQ Université Paris-Saclay, Sorbonne Université, CNRS, Guyancourt, France

**Correspondence:** Théophane Costabloz (theophane.costabloz@meteo.fr)

**Abstract.**

Observations collected during the SOuth west FOGs 3D experiment for processes study (SOFOG3D) field campaign are examined to document vertical profile of microphysical and thermodynamic properties of fog layers. In situ measurements collected under a tethered balloon provide 140 vertical profiles of liquid water content (LWC) from an adapted cloud droplet probe (CDP), which allow an exhaustive analysis of the life cycle of 8 thin fogs (thickness < 50 m) and 4 developed layers. We estimate thin-to-thick transition time from remote sensing instruments (microwave radiometer and Doppler cloud radar) and surface measurements, by using thresholds for longwave radiation flux, turbulent kinetic energy, vertical temperature gradient, fog top height and liquid water path (LWP) values. We found that a LWP threshold value of 15 $g.m^{-2}$ is more suited for the thick fogs sampled at the super-site. CDP data are used to compute the equivalent fog adiabaticity from closure ($\alpha_{eq}^{closure}$) and compare to value derived from remote sensing instruments, 2-m height visibility, and an one-column conceptual model of adiabatic continental fog assuming that LWC linearly increases with height. The comparison of $\alpha_{eq}^{closure}$ shows a large variability that results mainly from the parameterization used to estimate LWC at ground, but their evolution as a function of the fog thickness follows the same trend. We found larger negative values of $\alpha_{eq}^{closure}$ for thin layers, associated to low LWP values. CDP data reveal that reverse trend of LWC profile (LWC being maximal at the ground and decreasing with altitude) are ubiquitous in optically thin fogs, while quasi-adiabatic features with increasing LWC values with altitude are mainly observed in well-mixed optically thick fogs. We investigate the actual fog adiabaticity and lapse rate fraction by using linear regressions to best fit the vertical profiles of LWC and temperature, respectively. This analysis highligths that reverse LWC profiles, when stable temperature conditions exist during the optically thin phase of fogs, evolve towards quasi-adiabatic features with slightly unstable temperature lapse rate, when fogs become optically thick. We also found that LWC at ground is higher during the thin phase and significantly decreases as the profile is changing from reverse to increasing with height. But this trend could be balanced when collision-coalescence and sedimentation processes redistribute the LWC through the fog layer from the top to the ground. This study provides new insights on the evolution of LWC profile during the fog life cycle, that would help to constrain numerical simulations.



## 1   Introduction

Fog is defined by the World Meteorological Organization as water droplets (sometimes ice crystals) in suspension in the atmosphere reducing the visibility at the Earth's surface below 1000 m (5/8 mile) (World Meteorological Organization, 1956). This meteorological phenomenon affects human activities, and strongly perturbs the aviation, marine and land transportation, leading to human losses and high financial costs (Gultepe et al., 2007). Despite numerous studies on fog modelling, the accuracy of fog predictions by numerical weather prediction (NWP) models remains a challenge (Müller et al., 2010; Steeneveld et al., 2015; Boutle et al., 2018; Westerhuis et al., 2020)

The difficulties encountered are related to low horizontal (Bergot and Guedalia, 1994; Pagowski et al., 2004; Boutle et al., 2016) and vertical resolutions (Beare and Macvean, 2004; Tardif, 2007; Edwards, 2009), surface heterogeneities (Bergot et al., 2015; Mazoyer et al., 2017), surface conditions (Duynkerke, 1999), large-scale conditions (Koračin et al., 2001), and initial conditions (Rémy and Bergot, 2009). Fog life cycle results from a complex interaction between radiative cooling, turbulence, microphysics, and non-local effects. Roach et al. (1976) and Teixeira (1999) illustrated the impact of radiative cooling for reliable fog predictions. The role of turbulence (Musson-Genon, 1987; Turton and Brown, 1987) and non-local effects related to complex terrain (Müller et al., 2010; Cuxart and Jiménez, 2012; Ducongé et al., 2020) were also evidenced.

In particular, a main issue concerns the transition from optically thin-to-thick fog being too sudden in numerical simulations and forecasts due to an excessive amount of liquid water in the fog layer (Poku et al., 2021; Boutle et al., 2022; Antoine et al., 2023). A fog becomes optically thick during its development phase when the layer closest to the ground radiates sufficiently in the longwave (LW) range to warm the surface below (Mason, 1982; Price, 2011). The downward LW radiation then increases so that the net LW flux becomes zero (Duynkerke, 1999; Wærsted et al., 2017; Dupont et al., 2016; Dhangar et al., 2021), while the fog optical thickness increases (Vehil et al., 1989). Its geometric thickness also increases (Wærsted et al., 2017; Price, 2011), as does the liquid water path (LWP) which results from the contributions of geometric thickness and liquid water content (LWC). As the fog top rises, it begins to cool by LW radiation, while the lower part of the fog is shielded from cooling and tends to warm. These two effects destabilise the temperature profile (Roach et al., 1976; Price, 2011) and the vertical temperature gradient becomes negative near the ground (Dupont et al., 2016). This destabilisation in turn creates small vertical motions within the fog layer which gives rise to turbulence (Nakanishi, 2000). However, this transition from thin-to-thick fog is not systematic, contrary to what fog simulations usually predict. Observations at Cardington (UK) and during the LANFEX campaign (Price et al., 2018) have shown that only 50% of sampled events become optically thick fogs (Price, 2011, 2019). Performing sensitivity tests on droplet concentration, Boutle et al. (2018) and Ducongé et al. (2020) found an optically thin-to-thick fog transition more consistent with observations. They suggested that a lower droplet concentration leads to greater droplet sedimentation, resulting in lower LWC values in the fog layer and thus optically thinner fog. Numerous studies have shown that aerosol properties and droplet size distribution representations through microphysical scheme are also a major cause of uncertainty in fog simulation and forecasting (Bott, 1991; Zhang et al., 2014; Stolaki et al., 2015; Maalick et al.,



2016; Schwenkel and Maronga, 2019; Boutle et al., 2022; Fathalli et al., 2022). Therefore, observations of fog microphysics are essential to improve fog simulations.

Previous observations of ground-level microphysics have revealed large and rapid temporal variability of LWC in fogs
(Gerber, 1981; Choularton et al., 1981; Fuzzi et al., 1984). Fog campaigns also highlighted significant differences in droplet size distribution during fog episodes (Kunkel, 1984; Wendish et al., 1998; Gultepe et al., 2009; Niu et al., 2011; Price, 2011; Mazoyer et al., 2019), among many others). Recently, Mazoyer et al. (2022) examined the evolution of microphysics during the fog life cycle and showed that it depends on the vertical developpement of the fog layer. However, most fog campaigns were carried out at ground level or on low masts, and observations inside the fog layer are rare due to the difficulty of the
measurements. The pionnering experiments of Okita (1962) along the slope of the Mt. Tokaschi (2070 m A.G.L) and Pinnick et al. (1978) with a tethered balloon, provided the first measurements of vertical profiles of microphysical properties in fog. More recently, Okuda et al. (2010); Egli et al. (2015); Price et al. (2015) have also reported microphysical measurements using a tethered balloon. Most of these measurements were conducted in deep well-mixed mature fogs, or fogs that lifted into stratus. In general, they revealed LWC profiles that were roughly constant or increasing with height, similar to aircraft
measurements performed in stratus and stratocumulus clouds. Based on these observations and following the approach of Cermak and Bendix (2011), Toledo et al. (2021) have developed a one-column conceptual model of adiabatic continental fog by assuming that LWC linearly increases with height, but with a reduced condensation rate referred as the local adiabaticity. They used remote sensing data from 7 years of measurements performed at the SIRTA (Site Instrumental de Recherche par Télédétection Atmosphérique) observatory near Paris, to compute the equivalent local adiabaticity by closure that would give
the same LWP in the fog layer, but assuming a linear increase in LWC with height. They showed that this parameter is indeed positive for the majority of their data, corresponding to thick adiabatic and buoyant fog layers, but they noted some negative values for thinner fogs with LWP < 30 $g.m^{-2}$, suggesting that LWC could be higher at the surface and decrease with heigth in such cases. Using cloud radar reflectivity measurements, Wærsted et al. (2017) also retrieved higher LWC values near the ground for a thin fog event. But in situ observations of the vertical profile of fog microphysics are sorely lacking in the
literature.

The SOuth west FOGs 3D experiment for processes study (SOFOG3D) field campaign took place during winter 2019/2020 in the South-West of France to provide 3D mapping of the boundary layer during fog events (Burnet et al., 2020). The observation strategy combined vertical profiles derived from remote sensing instruments (microwave radiometer (MWR), Doppler cloud radar and Doppler lidars) and balloon-borne in-situ measurements of fog microphysics and thermodynamics. Bell et al.
(2022) and Vishwakarma et al. (2023) combined cloud radar reflectivity with temperature and humidity profiles and LWP retrieved from MWR, to better estimate the vertical profile of LWC in the fog layer. They demonstrated that LWC retrieval is highly sensitive to the prescribed droplet concentration, and that agreement with in situ data is highly dependent on cloud–fog heterogeneity. Dione et al. (2023) combined remote sensing measurements with the conceptual adiabatic fog model to analyze the thermodynamic and turbulent processes involved in fog formation, development and dissipation, focusing of the four
deepest case studies : two radiation fogs and two advection-radiation fogs. They defined the different phases characterizing the fog's life cycle and provided quantitative analyses of key paramemeters and conditions that drive their temporal evolutions.



In this study, we examine in situ microphysical measurements collected under the tethered balloon during the SOFOG3D field campaign, to document the vertical profiles of LWC in the fog layer. For the first time, these observations provide an exhaustive analysis of the evolution of vertical profiles of microphysical and thermodynamic properties during the fog life cycle, from the formation phase in a thin stable layer to the well-mixed fog layer once vertical development has occurred, and even during dissipation when fog lifted in a stratus cloud. They are used to investigate the actual fog adiabaticity in various case studies and to compare it with the equivalent value derived from the closure proposed by Toledo et al. (2021).

This article is organized as follows, Section 2 describes the dataset and proposes an estimate of the thin-to-thick transition time with an uncertainty period. Section 3 introduces fog adiabaticity, presents the methodology for analyzing the in situ data and compares these results with equivalent adiabaticity from closure values. Section 4 documents the evolution of LWC and temperature vertical profiles in the sampled fog layers and provides new information for both thin and optically thick fogs. These results are discussed in section 5, followed by the conclusion and outlook in section 6.

## 2 Dataset

### 2.1 Observationnal sites and instrumentation

The SOFOG3D experiment (Burnet et al., 2020) was conducted during the winter of 2019/2020 in the southwest of France in the Landes forest region (Fig. 1a). A total of 17 instrumented sites were spread over a 30 x 50 km area (red rectange in Fig. 1). The Jachère site (44.41° N, 0.61° W) has been selected in a fallow field located in a large open area (Thornton et al., 2023) and was specifically equipped for measurements of aerosols and fog microphysics as well as energy balance with in situ instruments at the surface and on masts. The Charbonnière site, 1.4 km away over a flat terrain (Fig. 1b), was specifically dedicated to remote sensing observations and tethered balloon operations. It was located close to an agricultural building for convenience, and was open from SW to NE with a small forested area on the other side. Measurements from these two sites will be analysed in this paper to document the evolution of the vertical profile of the microphysical properties.

The instruments used in this study are summarized in Table 1. Both sites were equipped with a ground-based meteorological station that provided standard dynamical and thermodynamical measurements such as temperature, wind speed and wind direction as well as longwave and shortwave radiation.

A 95 GHz BASTA cloud radar was operated at the Charbonnière site on a vertical pointing mode. It measured radar reflectivity and Doppler velocity up to 18 km with three vertical resolutions (12.5, 25, and 100 m) (Delanoë et al., 2016). The 12.5 m vertical resolution mode is dedicated to fog and low clouds, with the first available gate between 25 and 37.5 m a.g.l.. Cloud base height (CBH) and cloud top height (CTH) were provided at a time resolution of 3 s, by the level 2 product developed by combining the three modes to derive optimized radar reflectivity, velocity, and mask indicating the valid signal from noise. A additionnal mask has been defined to remove radar reflectivities when the tethered balloon was interfering with the cloud radar measurements (Delanoë, 2020). Since qualitative analysis of the BASTA reflectivities is performed here to provide geometrical thickness of the fog layer, we choose to use all the available data and this specific mask was not applied. An RPG HATPRO 1/10 Hz microwave radiometer (MWR) was also deployed at the Charbonnière site (Martinet et al., 2022). Using neural net-





a)
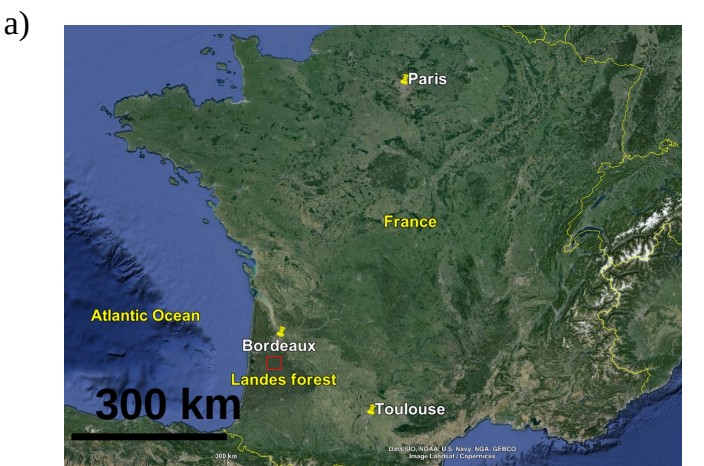

b)
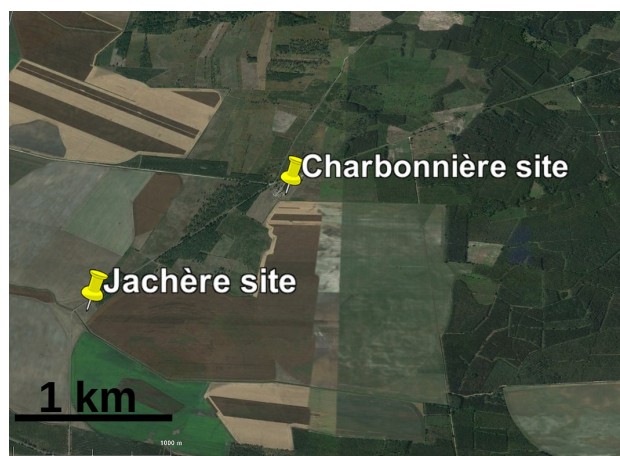

**Figure 1.** (a) Map of France with the location of SOFOG3D in red rectangle © Google Earth (b) Focus on the Jachère and Charbonnière sites © Google Earth.

work inversion, it provides vertical temperature and humidity profiles up to 2.5 km with a vertical resolution of 25 m up to 100 m high and 30 m above, as well as the liquid water path (LWP) over the whole layer. The synergy between both instruments was investigated for IOP 11 (Vishwakarma et al., 2023) and IOP 14 (Bell et al., 2022), by combining the LWP retrieved from the radiometer and the reflectivity from the BASTA radar in order to better estimate the vertical profile of LWC within the fog layer. However, we analyse here all the data collected during the SOFOG3D experiment, and we then use independant retrieval.

An 18 m$^3$ tethered balloon was operated at the Charbonnière site to provide in situ measurements through the boundary layer up to 500 m during the fog events. The payload consisted of an adapted DMT cloud droplet probe (CDP) for fog microphysics and meteorological sensors to provide thermodynamical vertical profiles of temperature, humidity, wind speed and wind direction. Depending on the period, a Vaisala tethersonde or a Gill ultrasonic anemometer and an inertial sensor were used for turbulence measurements (Canut et al., 2016), as indicated in Table 2. The CDP is an aircraft instrument that provides

the size distribution of cloud droplets from 2 to 50 μm in diameter at 1 Hz (Lance et al., 2010). To operate under a tethered balloon, a wind vane is used to align the sampling section perpendicular to the wind, and a small fan fixed just to the rear of the laser beam aspirates the air flow. The air speed in the sampling section is therefore equal to the wind speed plus 5 m.s$^{-1}$ (Fathalli et al., 2022).

## 2.2 Case studies

During the SOFOG3D campaign, 15 intensive observation periods (IOPs) with tethered balloon operations were conducted. Here we analyze 12 fog events sampled with the CDP under the balloon. Their charateristics are summarized in Table 2.



**Table 1.** List of instruments used in this study, references for uncertainty are : [1] (Bell et al., 2022), [2] (Martinet et al., 2022), [3] (Lance et al., 2010)

| Instruments | Measured Variable | Unit | Vertical Position | Uncertainty | Temporal Resolution |
| --- | --- | --- | --- | --- | --- |
| PT100 | Temperature | $^\circ$C | 2 m | 0.5 $^\circ$C | 60 s |
| Young Wind Monitor | Wind Speed | m/s | 10 m | 0.3 m/s | 60 s |
| | Wind Direction | $^\circ$ | 10 m | 3 $^\circ$ | 60 s |
| Kipp and Sonnen Spectrometer | Downward/Upward Longwave Radiation | W/m$^2$ | 1 m | 10 % | 60 s |
| METEK Sonic Anemometer | Turbulent Kinetic Energy (TKE) | m$^2$/s$^2$ | 3 m | 1.5 % | 30 min |
| Young Visibility Sensor | Horizontal Visibility | m | 3 m | 10 % | 60 s |
| Vaisala PWD22 | Horizontal Visibility | m | 3 m | 15 % | 15 s |
| BASTA Cloud radar | Reflectivity (Z) | dBZ | | 0.5-2 dBZ[1] | 3 s |
| | Cloud Base/Top Height (CBH/CTH) | m | | 12.5 m | 3 s |
| HATPRO Radiometer | Temperature | K | | 0.25 K RMS[2] | 10 min |
| | Liquid Water Path (LWP) | g/m$^2$ | | 20 g/m$^2$ | 1 s |
| XSENS Sensor | Altitude (Z) | m | | 5 m | 10 s |
| Rotronic Sensor | Temperature | $^\circ$C | | 0.1 K | 10 s |
| Gill Anemometer | Wind Speed | m/s | | 1.5 % | 10 s |
| Cloud Droplet Probe (CDP) | Liquid Water Content (LWC) | g/m$^3$ | | 30 %[3] | 1 s |
| Vaisala RS41-SGP | Temperaure | $^\circ$C | | 0.3 $^\circ$C | 1 s |

The time of formation and dissipation, and the type of fog are determined on the basis of measurements at the Jachère site since visibility measurements at the Charbonnière site were only available between January 23, 2020 and March 4, 2020. A fog event starts when visibilty falls below 1 km for at least 30 minutes, and dissipation occurs when visibility exceeds this

threshold for more than one hour.

The algorithm developed by Tardif and Rasmussen (2007) was implemented to determine the type of fog that should reflect the main processes leading to fog formation. It depends on the magnitude of radiative cooling and wind speed, as well as the presence of precipitation or stratus cloud prior to fog formation. But threshold effects appeared for several cases leading to fog being classified as either radiative or advective, even though these two major processes are equally important. This is

because Tardif and Rasmussen (2007) considered radiative-advective fogs to be radiative fogs because no distinction was made between heating and air drainage. Given the importance of the advective component observed during SOFOG3D linked to the proximity of the Atlantic Ocean, we analyzed large scale conditions using synthetic analysis products, in addition to local conditions in the Supersite's surroundings at the fog formation, using satellite data, radar and thermodynamical parameters from meteorological stations at the Jachère and Charbonnière sites. This analysis enabled us to determine the most appropriate

classification, allowing a fog to be classified as radiative-advective (Ryznar, 1977; Gultepe et al., 2007; Yang et al., 2018) if





**Table 2.** Summary of the IOPs from the SOFOG3D campaign used in this study, based on the measurements at the Jachère site (except for the transition time based on net longwave radiation from the Charbonnière site). The time interval between the first and last threshold (see text) is indicated in brackets. IOPs 13b, 14, 11 and 6c, will be studied more specifically in Section 4.

| IOP | Date | Formation ( UTC) Dissipation (UTC) | Duration | Type | Transition (UTC) (time interval) | Number of profiles : thin/thick/stratus - total |
|---|---|---|---|---|---|---|
| 2a | 05-06/12/2019 | 2139/ 0525 | 7h46 | Radiative | | 17/0/0 - 18 |
| 2b | 06-07/12/2019 | 1924/ 0514 | 5h32 | Radiative-Advective | | 26/0/0 - 25 |
| 6a | 03-04/01/2020 | 0141/ 0746 | 6h05 | Precipitations | | 11/0/0 -11 |
| 6b | 04-05/01/2020 | 2342/ 0121 | 1h31 | Radiative | | 3/0/0 - 3 |
| **6c** | 05-06/01/2020 | 2037/ 0928 | 12h51 | Radiative-Advective | 2126 (04h11) | 0/24/2 - 26 |
| **9a** | 23-24/01/2020 | 2103/ 0138 | 4h35 | Precipitation | 2104 (2h29) | 0/9/0 - 9 |
| 9b | 24-25/01/2020 | 0040/ 0256 | 2h16 | Radiative | | 3/0/0 - 3 |
| **11** | 08-09/02/2020 | 2038/ 0349 | 7h11 | Radiative | 0044 (0h25) | 4/4/4 - 12 |
| 13a | 22-23/02/2020 | 2303/ 0311 | 2h52 | Radiative-Advective | | 9/0/1 - 10 |
| 13b | 23-24/02/2020 | 2104/ 0018 | 3h14 | Radiative-Advective | | 8/0/2 - 10 |
| **14** | 07-08/03/2020 | 2120/ 0705 | 8h31 | Radiative-Advective | 0012 (0h48) | 3/3/4 - 10 |
| 15 | 11-12/03/2020 | 2242/ 0417 | 5h35 | Radiative | | 3/0/0 - 3 |

both aspects are considered significant. All other possible fog types were the same as those described by Tardif and Rasmussen (2007), i.e. radiative, advective, precipitation, stratus lowering and evaporation fogs.

Three deep fog events with CTH higher than 200 m were sampled with the tethered balloon, namely IOP-6c, 11 and 14. Given such vertical developpment they have clearly undergone a transition from thin to thick. The life cycle of these cases has been examined by Dione et al. (2023) to analyze the thermodynamics and turbulent processes involved in fog formation, evolution and dissipation. Most of the fog layers sampled at the Charbonnière site, however, reached much lower thickness than these 3 cases. Indeed we found only one additional thick fog on the database (IOP 9a), while the other 8 cases remained optically thin. For the 4 thick events, an estimation of the time of the thin-to-thick transition is provided in Table 2 with a duration reflecting the associated uncertainty based on a method described in the next section.

For the in situ measurements performed under the tethered balloon, the observation strategy consisted of ascents and descents through the layer to provide vertical profiles of the fog microphysics and thermodynamics, and constant-height sections at various altitude to investigate time evolution and turbulence within the fog layer. To be representative, the turbulent kinetic energy (TKE) is determined over constant-height sections lasting at least 20 minutes. Tethered balloon tracks for the four fog events analyzed in section 4 are illustrated in Fig. 8c, 9c, 10c and 11c superimposed to the radar reflectivity, for IOP 13b, 14, 11 and 6, respectively. The maximum ascent or descent speed of the balloon is $0.5 \, \mathrm{m.s^{-1}}$. It then theoretically takes 10 minutes





to cross over up and down a 150 m-thick layer. However, due to the increase of the wind with the altitude that tends to sweep off the balloon away from the winch, the required time is much higher. In addition, the CDP is powered with a battery that allows measurements up to a maximum of around 1.5 hr. As a result, some profiles did not cross the entire fog layer from the ground up to the CTH and were discarded for this study. For 5 episodes, some vertical profiles were performed after the fog

lifted into a stratus cloud during the dissipation phase, including the 2 thin cases sampled at the end of February. Overall a total of 140 vertical profiles has been selected, including 91 in thin fogs, 41 in thick fogs, and 14 in stratus clouds. The number of profiles is given for each IOP in the last column of Table 2. The number of profiles available in each phase of the life cycle is highly variable depending on fog duration, vertical extension and various technical difficulties encountered during operations. But with the exception of the 2 shortest events, there are between 9 and 27 profiles per fog case providing a unique dataset to

document the vertical structure of the thermodynamics and microphysical properties of fog layers.

### 2.3 Determination of the thin-to-thick transition time

As mentioned in section 1, a fog becomes optically thick when the layer closest to the ground radiates sufficiently in the longwave range to warm the surface below. This leads to the destabilization of the fog layer which evolves from a stable to a neutral or slightly unstable temperature profile. Precise estimation of the time of transition is not easy, however. Many authors

have proposed various thresholds for different radiative, thermodynamic, geometric and microphysical parameters.

We apply the following five conditions for the four thick fogs in order to assess the uncertainty associated with the definition of the transition time. To smooth out high frequency fluctuations that are not representative of the typical length scale of the fog phases, we compute the transition time for a given parameter from the 30-minutes running average of its time series except for TKE and vertical temperature gradient for which original sampling frequency data are used (Table 1).

– As the fog becomes optically thick, the longwave net radiation $LW_N = (LW_{DOWN} - LW_{UP})$ increases during the night and approaches 0 (Wærsted et al., 2017; Dupont et al., 2016; Dhangar et al., 2021). Mazoyer et al. (2017) observed a difference of 8 W.m$^{-2}$ between $LW_{DOWN}$ and $LW_{UP}$ after the fog vertical development. We consider that the transition is triggered when $|LW_N| < 5$ W.m$^{-2}$.

– Warming at the surface and cooling at the fog top destabilize the vertical profile of temperature, which reverses from

stable conditions and starts to decrease with height (Roach et al., 1976; Price, 2011). We use the temperature profile provided by the MWR just above the surface, but we have discarded the first two gates because of the excessive influence of the ground on the measurements. The considered threshold is when the temperature gradient between 50 m and 25 m becomes negative, i.e. $T_{50m} < T_{25m}$.

– Due to the destabilization of the vertical temperature profile, turbulent motions increase (Nakanishi, 2000). a threshold

on $\sigma_w^2$ values is difficult to define. Price (2019) proposed $\sigma_w^2$ values higher than 0.002-0.005 m$^2$.s$^{-2}$ at the transition. But Dione et al. (2023) found much larger values ($0.01 < \sigma_w^2 < 0.04$ m$^2$.s$^{-2}$) during the stable/adiabatic transition of the 4 deepest fog events of SOFOG3D and an increase in TKE of up to 0.4 m$^2$.s$^{-2}$. These discrepancies may be explained by the contrasting environment between the two measurement areas (Thornton et al., 2023). In the same way, Dhangar




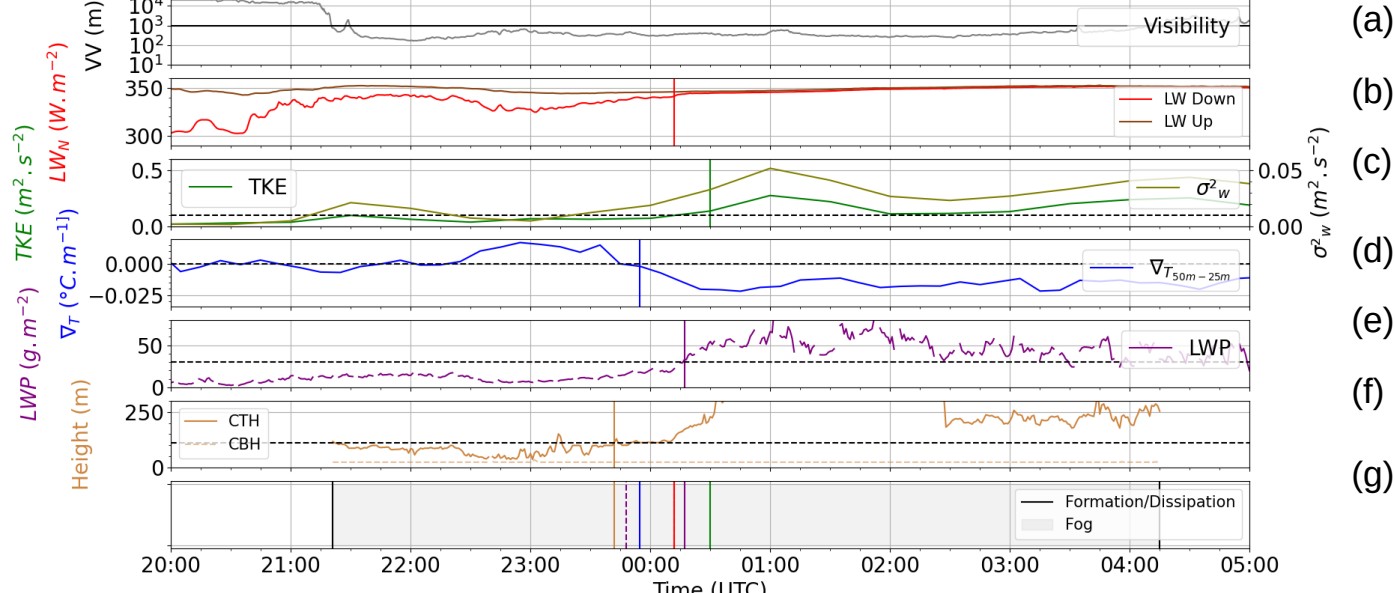

**Figure 2.** Time series for IOP 14 fog event : (a) visiblity (b) downward (light red) and upward (brown red) longwave radiative fluxes, (c) TKE (green) and $\sigma_w^2$ (light green), (d) vertical gradient of temperature between 25 and 50m from the MWR, (e) LWP from the MWR, (f) CTH and CBH from the BASTA cloud radar. In panels (c) to (f), threshold values are indicated by the dotted horizontal lines. In panels (b) to (f), the vertical segments represent the transition times. (g) Summary graph with the five transition times and their respective colours superimposed over the foggy period (grey area) delimited by the formation and dissipation times in vertical black lines. The vertical purple dotted segment represents the transition time derived from the LWP with a threshold of 15 $\mathrm{g.m^{-2}}$ instead of 30 $\mathrm{g.m^{-2}}$.

et al. (2021), during the WIFEX campaign over New Delhi, considered a TKE threshold of 0.10 $\mathrm{m^2.s^{-2}}$. We then choose TKE > 0.10 $\mathrm{m^2.s^{-2}}$.

– An increase in geometrical thickness is systematically observed as the fog becomes optically thick. This vertical development can be detected using the fog cloud top height derived from cloud radars (Wærsted et al., 2017) or tethered balloon systems (Price, 2011). Based on theses studies we consider that the transition occurs when CTH > 110 m.

– Finally, the LWP also increases. We apply the condition determined by Wærsted et al. (2017), i.e. LWP > 30 $\mathrm{g.m^{-2}}$.

Results for IOP 14 case study are illustrated in Fig. 2. Each transition time for a given parameter is represented by a vertical bar on the corresponding time series in Fig. 2b-f and they are all reported in Fig. 2g for a synthetic representation. As expected, the thin-to-thick transition is systematically associated with an increase in $LW_N$, TKE, LWP, CTH and a decrease in the vertical temperature gradient. For this case, the different transition time values are very close within a 48-min period around 0012 UTC. But there could be large discrepancies. Figure A1 presents the results for IOP 11 with transition times from 0030 UTC to 0055 UTC, except the 30 $\mathrm{g.m^{-2}}$ LWP threshold about two hours later. This threshold seems too high in this case





since the transition rather occurs when LWP reaches 15 g.m$^{-2}$. IOP 6 reveals the largest scattering (Fig. A2) with the LW flux exceeding 5 W.m$^{-2}$ very early (50 min) after the fog onset, about 2h30 before the TKE threshold, and 3h30 before both CTH and vertical temperature gradient thresholds. This appears to be mainly due to advective processes that perturbed the fog life cycle. Indeed all these parameters are also close to their respective thresholds (CTH = 87.5 m, TKE = 0.06 m$^2$.s$^{-2}$ and vertical

temperature gradient = 0.0007 °C.m$^{-1}$) but the vertical developpment of the fog layer is temporarily stopped as the surface wind decreased before its direction shifted from SW to SE (Fig. A2b). The fog started to deepen again around midnight and then all parameters reached their thresholds shortly after, except the 30 g.m$^{-2}$ LWP threshold which required 1h15 more. Finally, during IOP 9a (not shown), $LW_N$ indicates that the fog is already thick at its formation, due to the immediate condensation of liquid water over a 80 m thick layer, but with a rather low value of LWP < 5 $g.m^{-2}$. A cloud passing above temporarily caused

the fog layer to drop below 50 m, and as for IOP 6, other parameters then exceed their thresholds later on when the fog started to deepen again, resulting in a transition interval of 2h29 as indicated in Table 2.

For these four cases, the transition occurred with LWP values systematically much lower than the 30 g.m$^{-2}$ threshold. To be consistent with the other parameters, we find that a value of 15 g.m$^{-2}$ is more suited to detect the transition phase. The other estimations are in agreement during IOPs 11 and 14 during which there was a clear thin-to-thick transition, but they

are more dispersed during IOPs 9a and 6c since the fog became optically thick immediately after its formation or shortly after, respectively. However, these four cases do not reveal any trend in the order of exceeding the thresholds by the different parameters. Furthermore it appears that non-local processes such as the change in wind orientation or the advection of clouds can disrupt the usual fog life cycle by stopping the vertical development of the fog layer.

This analysis provides estimations of the transition time for each case with an associated period of uncertainty, independent

of the in situ measurements performed under the tethered balloon. These are now examined to document the evolution of the vertical profile of microphysical properties during the fog life cycle, with particular emphasis on fog adiabaticity.

## 3   Fog adiabaticity

### 3.1   Equivalent adiabaticity from closure

In adiabatic liquid clouds, the liquid water content increases almost linearly with the altitude as $LWC(z) = \Gamma_{ad}(T,P)(z - Zb)$,

where $Zb$ is the cloud base height and $\Gamma_{ad}(T,P)$ the condensation rate that depends on pressure $P$ and temperature $T$ at the cloud base (Betts, 1982; Albrecht et al., 1990; Brenguier, 1991) following:

$$\Gamma_{ad}(T,P) = \rho_d\left(\frac{(\epsilon + w_s)w_s l_v \Gamma_w}{R_d T^2} - \frac{g w_s P}{(P - e_s)R_d T}\right) \tag{1}$$

with $l_v$ the latent heat of vaporization, $g$ the acceleration of gravity, $R_d$ the dry air ideal gas constant, $\epsilon$ the ratio between the dry air to water vapor ideal gas constant equal to 0.622, $e_s$ the vapor saturation pressure, $w_s$ the saturation mixing ratio, $\rho_d$ the

dry air density, and $\Gamma_w$ is the moist adiabatic lapse rate given in Eq. 2 (Hummel and Kuhn, 1981):



$$\Gamma_w = \frac{g}{c_p} \frac{1 + \frac{l_v w_s}{R_d T}}{1 + \frac{\epsilon l_v^2 w_s}{R_d c_p T^2}} \tag{2}$$

with $c_p$ the specific heat of dry air at constant pressure.

The liquid water path of a cloud layer is the total amount of liquid water, as :

$$LWP = \int_{Zb}^{Zt} LWC(z)dz \tag{3}$$

where $Zt$ is the cloud top height. In shallow convective cloud where the condensation rate can be assumed relatively constant it follows: $LWP_{ad} = \frac{1}{2}\Gamma_{ad}(T,P)(Zt - Zb)^2$ (Albrecht et al., 1990).

Processes such as entrainment-mixing of dry air or precipitation formation, however, tend to reduce LWC values, and Betts (1982) introduced the in-cloud mixing parameter $\beta$ to reduce the condensation rate as $(1 - \beta)\Gamma_{ad}(T,P)$. Many studies have quantified departure from the adiabatic values using aircraft in situ data and revealed that the reduction in stratiform clouds

is much lower than in cumulus ones and results mainly from mixing at the cloud top and drizzle formation (Gerber, 1996; Wood, 2005; Brenguier et al., 2011; Braun et al., 2018). By using remote sensing instruments in shallow stratocumulus clouds, (Albrecht et al., 1990) also noticed a large reduction in LWP, compared to the adiabatic values, when drizzle is observed. Observations in fog are rare due to the difficulty of measurement, but previous studies have reported LWC profiles that are fairly constant or increase with height in a well-developed fog layer (Okita, 1962; Pinnick et al., 1978; Price et al., 2015; Egli

et al., 2015).

To retrieve the cloud base from satellite data, allowing discrimination between low stratus and fog, Cermak and Bendix (2011) developed a subadiabatic model of cloud microphysics. They derived a complex cloud profile parameterization used for LWP computation by dividing the cloud in three layers with different $\beta$ values to account for processes reducing the LWC near the ground, in the central region and at cloud top.

Recently, to improve nowcasting of fog dissipation, Toledo et al. (2021) developed a one-column conceptual model of adiabatic continental fog by assuming that the LWC linearly increases with height with a reduced condensation rate expressed as $\alpha(z)\Gamma_{ad}(T,P)$ where $\alpha(z)$ is the local adiabaticity. The LWP of a fog layer is then computed considering that the equivalent local adiabaticity $\alpha_{eq} = 1 - \beta$ remains constant throughout the fog layer from the ground to the CTH, as:

$$LWP = \frac{1}{2}\alpha_{eq}\Gamma_{ad}(T,P).CTH^2 + LWC_0.CTH \tag{4}$$

where $LWC_0$ is the LWC value at the ground.

Toledo et al. (2021) then used data from 7 years of measurements performed at the SIRTA (Site Instrumental de Recherche par Télédétection Atmosphérique) observatory near Paris, to compute this equivalent local adiabaticity $\alpha_{eq}$. Without measurements of the vertical profiles of LWC, they used an inversion of Eq. 4 to calculate $\alpha_{eq}$ by closure as :

$$\alpha_{eq}^{closure} = \frac{2(LWP - LWC_0.CTH)}{\Gamma_{ad}(T,P).CTH^2} \tag{5}$$



where LWP and CHT are provided by a HATPRO MWR and a BASTA radar, respectively, and $LWC_0$ is derived from the measured visibility at ground by the parameterization developed in Gultepe et al. (2006).

$\alpha_{eq}^{closure}$ is then the equivalent adiabaticity that would give the same LWP in the fog layer, but assuming a linear increase in LWC with height. Note that from Eq. 5, $\alpha_{eq}^{closure} = 0$ corresponds to a constant LWC profile equal to $LWC_0$ from the ground to CTH and is quite different from $\beta = 1$, which corresponds to the total evaporation of the cloud due to mixing with clear air

(Betts, 1982). They found that $\alpha_{eq}^{closure}$ depends mainly on the CTH and pointed out that thinner fog with LWP values lower than 20 g.m$^{-2}$ has $\alpha_{eq}$ values below 0.6 and can even reach negative values. Dione et al. (2023) used this model to analyze the four deepest fogs of SOFOG3D and revealed that $\alpha_{eq}^{closure}$ increases from 0 to 0.5 during the thin-to-thick transition and reaches 0.5 when LWP > 20 g.m$^{-2}$. We now use the CDP measurements to analyze fog adiabaticity derived from in situ measurements.

## 3.2    Vertical profiles of LWC and temperature from in situ measurement

Droplet size distribution recorded by the CDP under the therered balloon during SOFOG3D allows us to retrieve vertical profiles of LWC in the fog layer and then to examine the actual fog adiabaticity and compare to the equivalent values derived by closure.

Figure 3 presents vertical profiles of LWC and temperature measurements collected during IOP 14 (upper panels) and IOP

11 (lower panels). Boxplots correspond to statistics computed within 5 m height layers from the ground up to the fog top. Black lines indicate the adiabatic theoretical calculation of LWC and lapse rate, from Eq. 1 and 2 respectively.

The ascent of IOP 14 was performed between 0611 and 0647 UTC, about 6 hours after the fog became optically thick (Fig. 2) and 2 hours before its dissipation in stratus (profile # 6 of Fig. 9d). As a general trend, LWC values increase with the altitude up to 215 m, before they drop suddenly in the upper fog layers near the fog top at 255 m (Fig. 3a). In this deep fog, however,

we can observe that the increase of LWC is not continuously monotonic, with the presence of a layer with much lower LWC values at heights between 120 and 170 m. The vertical profile of temperature decreases almost regularly with height up to the fog top (Fig. 3b). Unfortunately, the balloon failed to cross the summit due to the increase in wind speed to over 6 m.s$^{-1}$, but the radiosonde launched 35 minutes earlier indicates a sharp temperature inversion of -1.5 °C (not shown). These observations are consistent with previous measurements in well- mixed fog layer revealing mainly adiabatic vertical profiles of LWC and

temperature, but also a sharp reduction in LWC at the fog top probably due to entrainement-mixing of dryer air in the upper fog layers.

In contrast, the descent during IOP 11 between 2209 and 2223 UTC reveals a reverse trend with LWC being maximal at the ground, around 0.11 g.m$^{-3}$, and decreasing with altitude up to the CTH at 55 m, except a thin slice around 40 m. It is associated with a stable vertical profile of temperature that increases from 8.5 °C at ground to 10.1 °C above the CTH. Indeed

at this time (profile #2 of Fig. 10d) the fog is still optically thin as the transition to thick fog will occur about 1 hour and a half later. Such reverse LWC profiles were almost systematically observed in optically thin fogs during the SOFOG3D campaign and are investigated in more details in section 4. These results hightlight that while the adiabatic model correctly represents



**Figure 3.** Adiabaticity (left) and lapse rate (right) from CDP vertical profiles collected during (a and b) a descent of IOP 14 between 0611 UTC and 0647 UTC and (c and d) a descent of IOP 11 between 2209 UTC and 2223 UTC. Boxplots are derived from CDP data collected within 5 m height layer. The theoretical and local values are indicated by a black and red line respectively. The equivalent adiabaticity retrieved by remote sensing ($\alpha_{eq}^{closure}$) and in situ measurements ($\alpha_{eqCDP}^{closure}$) are indicated by a dashed and dash-dotted blue line respectively.



thermodynamical and microphysical properties of well-mixed fogs, it does not represent the properties of optically thin fogs at all.

For LWC profiles, the linear increase corresponding to $\alpha_{eq}^{closure}$ calculated from Eq. 5 following Toledo et al. (2021) is plotted in dashed blue line with values of 0.45 and 0.53 for IOP14 and IOP11, respectively. As expected for the well-mixed fog case, the resulting profile of LWC tends to underestimate actual values of LWC in the lower part of the layer with predicted values that can be half the measured ones. The dotted and long dashed blue line corresponds to the fog adiabaticity from closure but computed with the CDP data $\alpha_{eqCDP}^{closure}$ (see next section), and produces a very similar result. Indeed, these low values of

$\alpha_{eq}^{closure}$ result mainly from the strong reduction of LWC at the fog top. Consequently, the corresponding profiles of $\alpha_{eq}^{closure}$ are not really representative of the global agreement of the measured LWC with the adiabatic model. This overall agreement is, however, clearly reflected on the temperature adiabatic (Fig. 3b, black) and local lapse rates (Fig. 3b, blue), the latter being less affected by the evaporation following the mixing below the fog top. Surprisingly, the reverse LWC profile of IOP 11 is well reproduced with data aligned along the negative slope -1.04 of $\alpha_{eqCDP}^{closure}$. In contrast for this case, $\alpha_{eq}^{closure}$ is positive due

to a larger LWP value provided by the MWR (7.12 g.m$^{-2}$) compared the CDP measurements (2.81 g.m$^{-2}$) and the profile is shifted to larger LWC due to an excessive value of $LWC_0$ provided by the parameterization of Gultepe et al. (2006) (0.17 g.m$^{-3}$) compared to the CDP measurements (0.11 g.m$^{-3}$).

To better estimate the agreement of the measurements with the adiabatic model, we calculate the local adiabaticity $\alpha$ and $\gamma$ with respect to LWC and lapse rate, respectively, by determining the slope of the linear regression between medians values of

the statistics at each altitude range. To avoid underestimation of local adiabaticity resulting from entrainment-mixing processes below the fog top for well-mixed fog, we discard this layer by truncating the vertical profile at the altitude where LWC reaches its maximum value. The corresponding profiles are superimposed as red lines on Fig. 3a and b. With such a procedure $\alpha = 0.64$ for IOP 14 and the corresponding fit to the measurements better represents the general shape of the vertical profile. For the lapse rate $\gamma = 0.97$ which is then very close to the adiabatic decrease. In contrast, the descent during IOP 11 reveals negative $\alpha$

and $\gamma$ values of -0.82 and -7.26 respectively.

In a first step, we take advantage of the 140 selected profiles to evaluate the agreement between $\alpha_{eq}^{closure}$ and the equivalent values derived from in situ data.

### 3.3   Comparison between equivalent fog adiabaticity from closure estimations

We use the CDP data to compute the fog adiabaticity from closure $\alpha_{eqCDP}^{closure}$ from Eq. 5: CTH is determined by using an LWC

threshold of 0.01 g.m$^{-3}$, the LWP is calculated by integrating the LWC in each altitude range up to CTH, and $LWC_0$ is defined as the median value of LWC in the lowest altitude range. The median height corresponding to $LWC_0$ is 3.5 m with quantiles 25 % and 75 % of 2.5 m and 6.2 m respectively, i.e. very close to the height of the visibilimeters deployed at 3 m above ground level.

Comparison of $\alpha_{eqCDP}^{closure}$ vs. $\alpha_{eq}^{closure}$ are reported on Fig. 4a and reveal large discrepancies, with very low values of the

coefficient of determination $R^2$ and the slope of the linear regression which reach 0.16 and 0.29, respectively. The first available gate of the radar being 37.5 m, it obviously can not detect CTH below this height. In total 21 profiles performed by the tethered



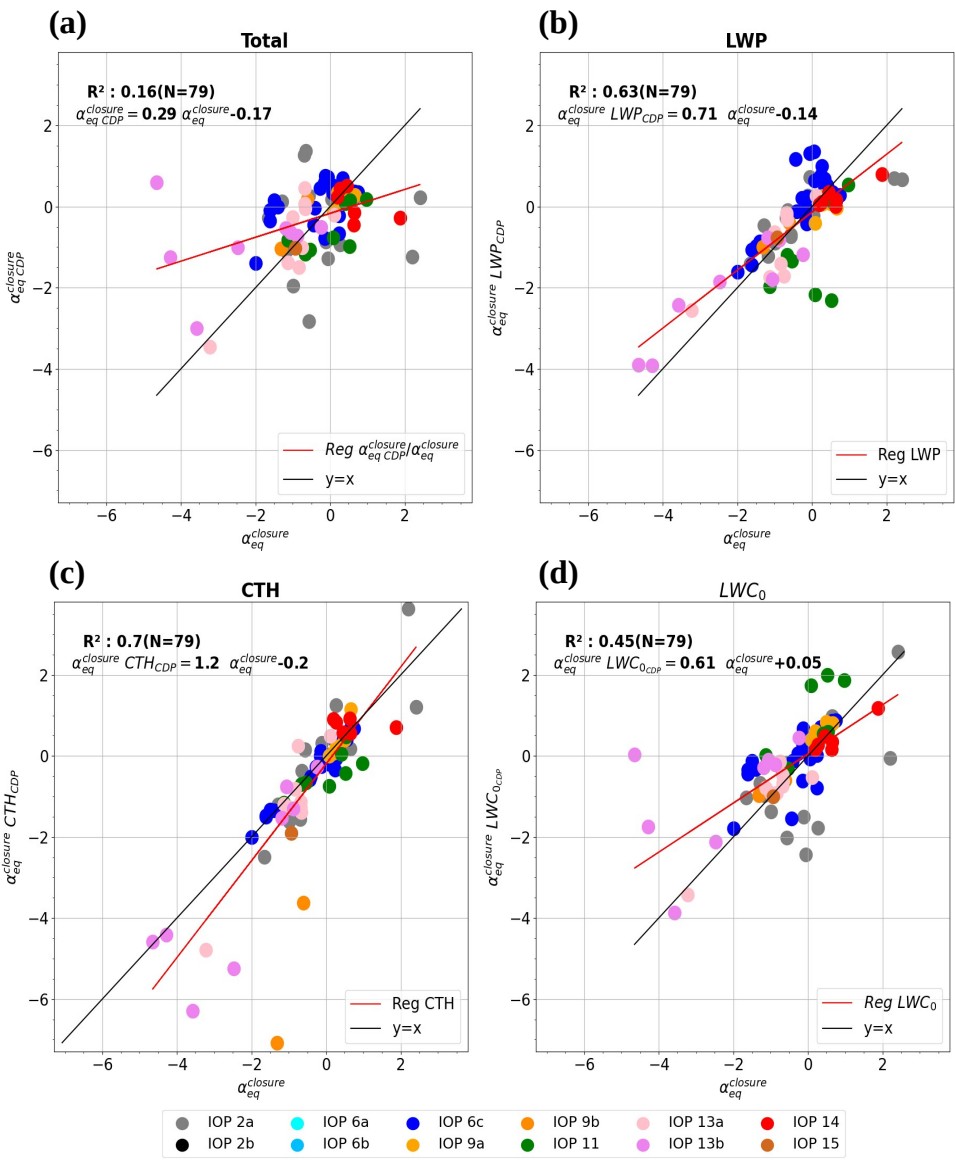

**Figure 4.** Fog adiabaticity from closure values comparison : (a) $\alpha_{eq}^{closure}$ values derived from the CDP measurements as a function of $\alpha_{eq}^{closure}$. Sensitivity tests performed on $\alpha_{eq}^{closure}$ when using only one parameter derived from CDP measurements: (b) LWP, (c) CTH, and (d) $LWC_0$. Colour of dot depends on IOP as indicated in the legend.

balloon are lower than this limit, that is about 20 % of the data set. This provides an additional motivation to compute $\alpha_{eqCDP}^{closure}$ in order to evaluate the conceptual approach against the actual vertical profile for the thin layers which correspond to the formation phases of many radiation fog events in stable conditions. For LWP retrieved from the MWR measurements, caution should be taken when clouds above the fog layer were detected, because it is not possible to dissociate the LWP values of the



fog layer and of the cloud layer above. Therefore, due to the presence of clouds above fog during IOP 2b and IOP 6a, these two cases were discarded. Finally, profiles within stratus cloud have been also removed to allow calculation of $\alpha_{eq}^{closure}$. In total the dataset is then reduced to 79 profiles. Note that both methods lead to values of $\alpha_{eq}^{closure} > 1$ but not for the same cases. This suggests that such values result from inaccuracy in measurements rather than reflecting superadiabatic conditions.

Sensitivity tests are carried out on LWP, CTH and $LWC_0$, by modifying only one of the 3 variables in Eq. 5 using the CDP measurements, the other two remaining based on $\alpha_{eq}^{closure}$ calculation in order to determine which parameters introduce the most dispersion in the comparison. Discrepancies seem less related to LWP and CTH as the sensitivity tests on these two variables present satisfactory correlations with $R^2$ values around 0.7 for both parameters (Fig. 4b-c). Indeed comparison of LWP provided by the MWR compared to the CDP measurements reveals a very good agreement, including for values below

15 g.m$^{-2}$ (Martinet et al., 2024) (In prep). Nevertheless, the worst cases on Fig. 4b (2 green points) corresponds to the IOP11 profiles with LWP from CDP much lower than the MWR value and highlights the impact of such difference. For CTH, the worst cases, from IOP 9b and 13b are mainly due to overestimation of CTH by the radar for actual CTH just above its detection limit that results in overestimation of $\alpha_{eq}^{closure}$. Finally, the strongest differences are observed when $LWC_0$ is replaced by the CDP values: Fig. 4d shows that there is a strong increase of the scatter of the data, consistent with a poor value of the coefficient

of determination of 0.45. Indeed, the comparison of $LWC_0$ values is reported in Fig. 5 which reveals large discrepancies with a correlation coefficient as low as 0.44. This confirms that the 2 worst cases from IOP 13b (pink points) on the comparisons of Fig. 4a and Fig. 4d, which correspond to $\alpha_{eq}^{closure} < -4$ whereas $\alpha_{eqCDP}^{closure}$ reaches -1.8 and 0, are due to $LWC_0$ from the parameterization that are about twice the values measured by CDP. The slope of the regression is 0.59 but this likely results from the large scatter of the data and it is not clear that the parameterization of Gultepe et al. (2006) with visibility measurements

tends to produce systematically LWC values larger than the CDP measurements. Therefore, these sensitivity tests reveal that most of the discrepancies between $\alpha_{eq}^{closure}$ and $\alpha_{eqCDP}^{closure}$ arise from the $LWC_0$ values derived from visibility measurement through the parameterization of Gultepe et al. (2006). Note however, that the data on Fig. 4a are not completely distributed in the same way as in Fig. 4d, reflecting that some differences between the parameters compensate for each other.

Toledo et al. (2021) have shown that $\alpha_{eq}^{closure}$ statistically increases with both CTH and LWP from negative values for thinner

fogs to values > 0.5 and converging to $\approx 0.7$ for developed fogs. They further proposed a parametrisation as a function of fog thickness (their Eq. (7)) to perform conceptual model calculations. Dione et al. (2023) obtained similar results to those of Toledo et al. (2021) for the four deepest fogs of SOFOG3D but with a lower LWP threshold of 20 g.m$^{-2}$ for $\alpha_{eq}^{closure} > 0.5$. We therefore examined evolution of $\alpha_{eqCDP}^{closure}$ as function of $LWP$ and $CTH$ issued from the CDP measurements for our dataset in Fig. 6 where measurements in stratus clouds have been excluded for consistency with previous studies. The observations

reveal similar trend with mainly negative values of $\alpha_{eqCDP}^{closure}$ when LWP < 5 g.m$^{-2}$, which represent 67 % of the sampled profiles. Note that some values reach -4, much lower than those reported in Toledo et al. (2021). Indeed they correspond to very thin layers with CTH in the blind zone of the radar BASTA. Except for two profiles of IOP 11 (green dots), $\alpha_{eqCDP}^{closure}$ is always positive when LWP exceeds 10 g.m$^{-2}$. Surprisingly, these two profiles of IOP 11 correspond to significant values of CTH (Fig. 6b). Indeed we will see in section 4.2.2 that these particular reverse profiles of LWC result from the sedimentation

process. Beyond a LWP of 15 g.m$^{-2}$, $\alpha_{eqCDP}^{closure}$ tends to converge around 0.5, but the number of samples is too limited for



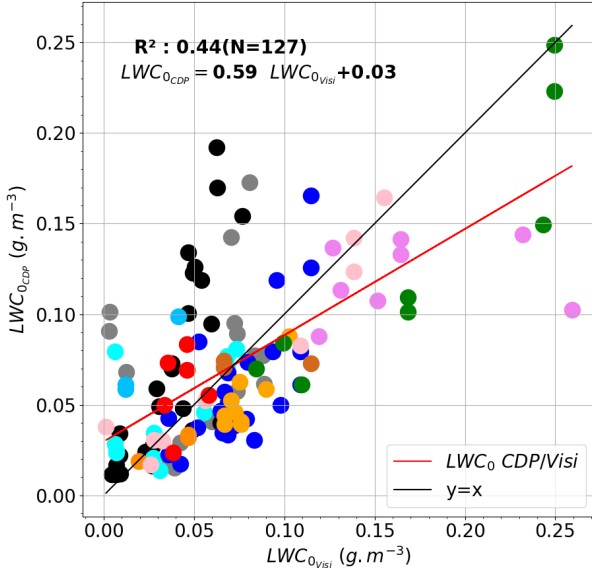

**Figure 5.** $LWC_0$ derived from the CDP measurements as a function of $LWC_0$ derived from the visibility measurements at the Jachère site using the parameterisation defined in Gultepe et al. (2006) Colours of dot corresponding to IOP are given in Fig. 4.

a precise evaluation of this limit. Two cases from IOP 2a exhibit superadiabatic conditions with $\alpha_{eqCDP}^{closure} > 1$. In fact, these profiles correspond to a particular case where the condensation of liquid water at the fog formation occurred first at altitude before the surface which therefore distorts the slope calculation (see Fig. C1).

The evolution of $\alpha_{eqCDP}^{closure}$ as a function of CTH depicted in Fig. 6b is very similar. The parametrisation of Toledo et al. (2021) has been defined for CTH > 85 m corresponding to the first gate of the radar used at the SIRTA site. This limit has been extended here to 10 m (dashed line) to compare with in situ data : most of them are situated above this curve reflecting that $\alpha_{eqCDP}^{closure}$ becomes positive for CTH lower than 100 m. Statistics on classes of 10 m width indicate that the median of $\alpha_{eqCDP}^{closure}$ values is positive when CTH > 60 m. For CTH > 150 m, $\alpha_{eqCDP}^{closure}$ values fluctuate around 0.4. This limit is lower than the convergence values of 0.65 and 0.6 reported for fog conditions in Toledo et al. (2021) and Dione et al. (2023), respectively. However, the number of profiles sampled in such developped fogs is obviously too limited here to derive significant convergence. Note that calculations of $\alpha_{eq}^{closure}$ with remote sensing measurements for the same profiles exhibit the same trend and do not improve the convergence estimation (Fig. B1). This attests that differences observed from our dataset result from the actual properties of the fog sampled during SOFOG3D and not from the measurements used (in situ or remote sensing) to compute the fog adiabaticity from closure.

In summary, the comparison of equivalent fog adiabaticity from closure calculations reveals that, despite the large variability that results mainly from $LWC_0$ and LWP retrievals, the distribution of equivalent fog adiabaticity is approximately the same between in situ or remote sensing measurements and the Toledo et al. (2021) approach. Additionnally, in situ data allow to



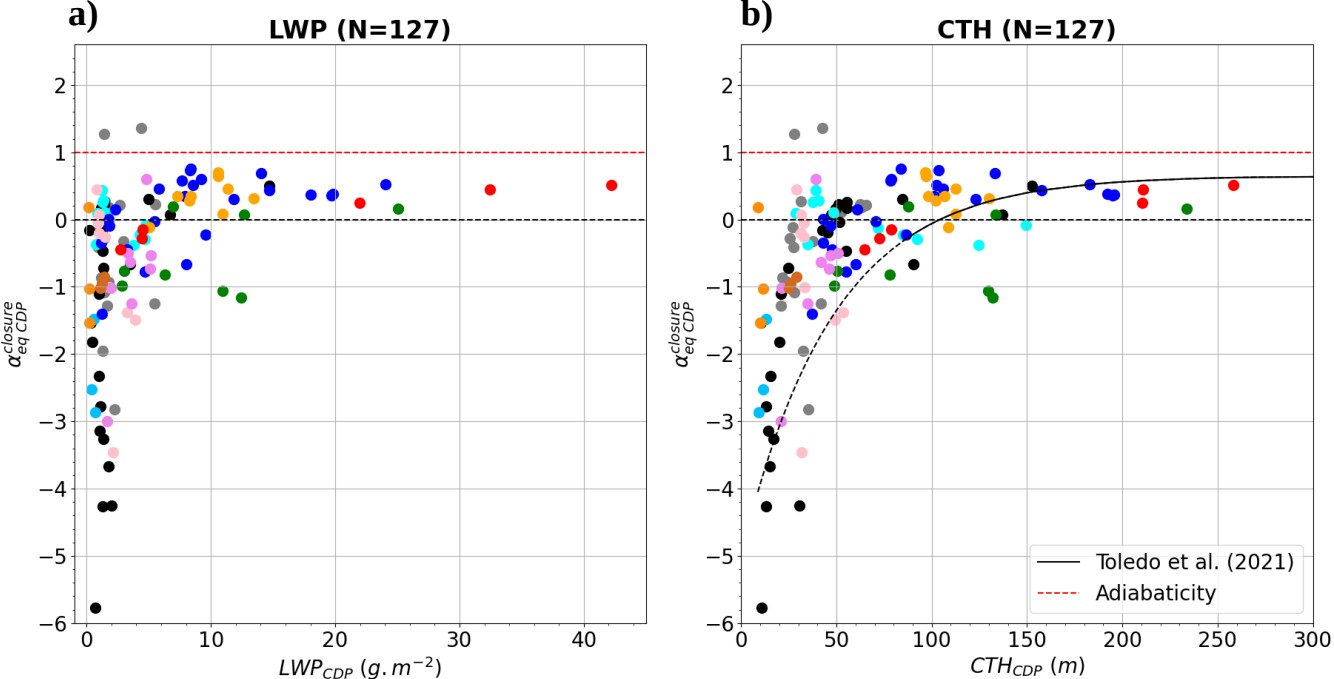

**Figure 6.** Adiabaticity fraction as a function of (a) LWP and (b) CTH derived from the CDP data for the 12 IOPs. Each IOP is indicated with a specific colour from IOP 2a to IOP 15 given in Fig. 4. The parametrisation between $\alpha_{eq}^{closure}$, and CTH defined by Toledo et al. (2021) is indicated by a black curve and the red line marks the adiabaticity.

characterize large negative $\alpha_{eq}^{closure}$ values relative to thin fogs. We now focus on in situ CDP data to examine relationships of equivalent fog adiabaticity with the real local adiabaticity.

## 3.4 Comparison between $\alpha$ and $\alpha_{eq}^{closure}$

We introduced at section 3.2 a calculation of the local adiabaticity $\alpha$ by fitting CDP measurements (red lines in Fig. 3a,c). The diluted layer just below the top is not taken into account to represent the general shape of the LWC vertical profiles. The main objective is now to derive a synthetic parameter to better quantify departure from the adiabatic model and document its evolution during the fog life cycle. To perform a quantitative evaluation of this parameter, we extend the analysis to the 140 selected profiles of the campaign. The thermodynamical and microphysical properties of each vertical profile are calculated with a vertical resolution of 5 m for episodes that experienced thin-to-thick transition. For fogs that remained thin and very thin (less than 10 m) throughout their life cycle, the vertical resolution is increased to 3 m and 1 m, respectively.With the exception of very thin fog episodes, a threshold for the minimum number of observations over each altitude range was set at 5. Given that, the average speed of the tethered balloon is about 0.5 m.s$^{-1}$, this represents a height of about 3 m corresponding to the vertical resolution required for most vertical profiles. Then, $\alpha$ and $\gamma$ are calculated only on representative vertical profiles





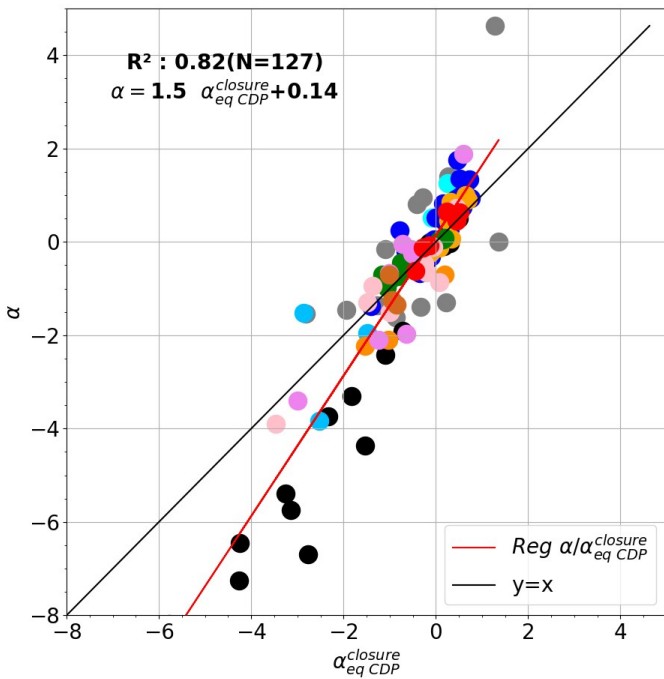

**Figure 7.** Adiabaticity fraction $\alpha$ as a function of the equivalent adiabaticity issued from the CDP measurements. Each IOP is indicated with a specific colour from IOP 2a to IOP 15 given in Fig. 4.

gathering at least 5 different altitude ranges. As discussed above, the diluted region below CTH is removed for the well-mixed fog. For doing so, $\alpha$ is first calculated over the entire profile. Cases with $\alpha > 0.1$ are identified as increasing with height and $\alpha$ is therefore recalculated on the lower part of the profile delimited by the maximum median LWC values, otherwise there are classified as reverse profile.

Comparison of $\alpha$ with $\alpha_{eqCDP}^{closure}$ are reported on Fig. 7 for the 127 selected profiles. As expected, the absolute values of $\alpha$ are generally larger than those of $\alpha_{eqCDP}^{closure}$ as attested by the slope of linear regression of 1.51. Some values of $\alpha$ exceed 1 but the examination of such profiles does not reveal any superadiabatic growth. In fact, these values rather result from the uncertainties of the method, which could be very sensitive to the highest values of LWC below the top or $LWC_0$ at the bottom (see profiles # 6 and # 7 of IOP 6 on Fig. 11 for example). This is particularly the case for the first profile of IOP2a (in grey), where high

values of LWC > 0.1 g.m$^{-3}$ at 20 m height while condensation at the surface has not yet taken place, lead to an $\alpha$ as high as 4.63 as depicted in Fig. C1c. Fitting the CDP data also produces higher negative values for reverse LWC profiles. Appendix C1a illustrates the corresponding profile for one of the lowest values ($\alpha$=-8.3) obtained during IOP 2b, which appears to be realistic given the linear regression on the vertical profile of LWC. Note that for such thin fog layer of 10 m height, the LWP is only 0.7 g.m$^{-2}$ although $LWC_0$ reaches 0.1 g.m$^{-3}$.



Such reverse profiles were frequently observed during the experiment and are mainly associated with stable atmospheric conditions resulting from radiative cooling established before the fog formation phase. As a next step, the evolution of vertical profiles during the fog life cycle is now studied in more detail to determine to what extent LWC profiles correlate with temperature profiles and how they both evolve during the thin-to-thick transition, when it exists, towards the adiabatic shapes of well-mixed fog layers.

## 4 Evolution of vertical profile properties during the fog life cycle

The analysis is first carried out on an episode that remains optically thin throughout its life cycle as the majority of fog cases sampled during the SOFOG3D campaign (8 out of 12, Table 2). Then, the contrast with fogs that underwent a transition from optically thin-to-thick is investigated with three other case studies. Finally, a generalization of the thermodynamical and microphysical properties over all fog samples of the 12 IOPs is conducted in section 4.3.

### 4.1 Thin fog event : IOP 13b case study

The radiation fog sampled during IOP 13b occurred during the night of February 23-24, 2020, associated with anticyclonic conditions and clear sky during the day, leading to a significant radiative cooling shortly after sunset, while wind speed remained low during this period (Fig. 8a). The fog appeared in patches around 2000 UTC and remained intermittent for two hours before visibility dropped below 1 km for a longer period between 2200 and 0000 UTC (Fig. 8a). At this point, radiative cooling stopped and the temperature stabilized at around 2.5 °C until the fog dissipated at 0018 UTC due to the advection of a 300-m thick stratus cloud at 2330 UTC, as depicted by radar reflectivity (Fig. 8c). It should be noted that this fog episode is too thin to be detected by radar, with the exception of a few isolated signals between 25 and 50 m height.

The eight profiles performed in the fog layer are displayed in Fig. 8c-d for temperature and LWC, respectively. Despite an increase of the CTH from 20 to 50 m, this fog remains optically thin throughout its life cycle (Fig. 8b). This optical thinness is associated with strong stable vertical profiles of temperature throughout the night, with $\gamma$ values ranging from -34.4 to -14.5. LWC vertical profiles are characterized by high values of LWC near the ground of around 0.1-0.2 g.m$^{-3}$ which tend to decrease with height, particularly profiles # 2 and # 3 (Fig. 8d) or to remain almost constant. Consequently, $\alpha$ values are also negative throughout the fog life cycle, ranging from -3.4 to -0.04, with the exception of vertical profile # 4 which reaches 1.89 due to high variability of LWC near the ground. Figure 8f illustrates the temporal evolution of $\gamma$ and $\alpha$ during the fog life cycle and clearly shows that both parameters increase progressively with CTH from negative values close to 0, and become positive in the stratus cloud after fog dissipation. Therefore, during a fog episode that remains optically thin throughout its life cycle, stable conditions persist and appear to be associated with a vertical profile of LWC more or less decreasing with height, in contrast to the adiabatic characteristics usually observed in well-developped fogs.





**Figure 8.** Temporal evolution during IOP 13b of (a) visibility, temperature at 2m and wind barbs at 10 m above ground level (empty circles, half barbs and full barbs represent wind speeds < 2.5, 5 and 10 m.s$^{-1}$, respectively. The orientation of the barbs gives the wind direction); (b) the upward (red) and downward (brown red) longwave radiation. The gray shaded area delimitate foggy periods; (c) reflectivity, CTH and CBH derived from the BASTA cloud radar. The trajectory of the tethered balloon is superimposed in grey. Each selected vertical profile is highlighted in blue and labeled by its number; (d) Corresponding vertical profiles of measured temperature, with local adiabatic (red) and moist adiabatic lapse rate (black); (e) Corresponding vertical profiles of LWC measured by the CDP with local (red) and theoretical (black) adiabaticity; (f) Temporal evolution of the adiabaticity (blue line) and lapse rate (green line) for the selected profiles.





## 4.2 Fog events with thin-to-thick transition

### 4.2.1 IOP 14 case study

IOP 14 took place on the night of March 7-8, 2020, characterized by a ridge. Shortly after the last residues of the warm front had dissipated, radiative cooling set in over at the Charbonnière site associated with a low wind (Fig. 9a). A low cloud appeared on the reflectivity at around 150 m height at 2030 UTC (Fig. 9c) and lowered for 40 minutes before reaching the ground at 2120 UTC as fog formed. This cloud formation at 150 m height may be explained by the presence of a fence of trees near the Charbonnière site, i.e. in the direction of the wind, as simulated on the SIRTA site by Mazoyer et al. (2017) using Large Eddy Simulations. Due to radiative and advective processes, this fog was classified as a radiative-advective case as Dione et al. (2023) and not as a fog by stratus lowering due to its rapid descent. The fog layer remained about 100 m thick for one hour but the CTH then rapidly lowered as the wind speed aloft dropped, with the wind shifting from SW to SE direction between 2230 and 2300 UTC (see Fig. 10d of Dione et al. (2023)). The ground temperature then decreased until 2330 UTC (Fig. 9b) when the fog layer began to thicken again.

The thin-to-thick transition times determined from the different parameters are very close within a 48-min period around 0012 (Fig. 9c and Fig. 2 in section 2.3). In fact, for this case, the advection around 0030 UTC of a stratus with a base height just above the fog top height contributes to the increase of LWP above the 30 g.m$^{-2}$ threshold. The CTH derived from the radar also increased sharply above 400 m, but in situ measurements from the tethered balloon attest that it is a distinct cloud layer above the fog. This transition period is included in the period determined by Dione et al. (2023) which extends from 2330 to 0100 UTC but is twice as short. The fog began to dissipate at 0416 UTC but visibility remains around 1000-2000 m for 2 hours before it dissipated completely at 0705 UTC, shortly after sunrise, and lifted in stratus cloud that persisted until 1000 UTC.

The first three vertical profiles with CTH < 80 m reported in Fig. 9d-e were performed during the optically thin phase of the fog event. The first profile reveals low negative values of -0.62 and -1.46 for $\alpha$ and $\gamma$, respectively, with a reverse LWC profile which rapidly became almost constant with height for profiles # 2 and # 3 with $\alpha$ values close to 0. Vertical profiles # 4 and # 5, which are consecutive descent and ascent after the transition, show contrasting microphysical properties. Profile # 4 indicates a continuous increase of LWC with height throughout the fog layer, except at the fog top, consistent with the quasi-adiabatic characteristic. But the $\alpha$ value only reached 0.46, indicating that some dilution has occurred. The vertical profile # 5 exhibits a similar trend up to a height of 100 m, but above that, in contrast, the LWC gradually decreased up to the CTH. The temperature profiles show a significant warming between 100 and 200 m height (+ 2 °C) which explains the evaporation of liquid water in the upper part of the layer. It is likely that this warming is linked to the advection of a stratus above the fog top. In this case, the $\alpha$ value reached 0.65, but this represents only the lower part of the profile. The last vertical profile performed in the fog layer 5 hours later depicts also an adiabatic profile, as discussed previously (Fig. 3).

Finally, the three profiles in the stratus following fog dissipation indicate a similar trend with $\alpha$ and $\gamma$ values between 0.56-0.69 and 0.84-1.0, respectively. Figure 9f illustrates the temporal evolution of $\alpha$ and $\gamma$ during the fog life cycle, with the





**Figure 9.** Same as Fig. 8 but for IOP 14. In addition b) the different thin-to-thick fog transition times are indicated by vertical lines as in Fig. A1; c) and f) the transition time for the net longwave radiation is indicated by a vertical red dashed line.



thin-to-thick transition indicated by the dashed segment. It appears that $\alpha$ and $\gamma$ distinctly transition from negative values, when the fog is optically thin, to positive values after the fog development.

### 4.2.2 IOP 11 case study

IOP 11 took place on the night of February 8-9, 2020, characterized by the establishment of a ridge after an inactive cold front had crossed the country during the day. Shortly after sunset, the sky cleared and radiative cooling set in over the "Supersite", leading to the formation of a radiative fog at 2039 UTC (Fig. 10b). This fog persisted until 0349 UTC when advection of warm air at the supersite caused the fog to lift into low stratus, which gradually dissipated until 0830 UTC.

As pointed out in section 2.3, the thin-to-thick transition time values are very similar for the different parameters between 495 0030 and 0055 UTC except for LWP which took two hours more to exceed the 30 g.m$^{-2}$ threshold (Fig. A1 in Appendix). Dione et al. (2023) considered that the transition period extended from 2300 to 0230 UTC which is the longest period of their 4 cases studies.

Of the nine selected vertical profiles reported in Fig. 10d-e, four were performed during the thin phase, one during the transition, three during the thick phase and the last shortly after the fog lifted into stratus. As with IOP 14, the first three 500 vertical profiles during the thin phase with CTH < 80 m present stable lapse rate and reverse LWC profiles with the highest values near the ground. Both $\alpha$ and $\gamma$ values are negative but the $\gamma$ values range from -2.45 and -8.97, which are higher, as for IOP 14, than those obtained during thin fog at IOP 13b, where they are systematically lower than -15. Consistently, $\alpha$ values range from -0.44 to -0.8, which are also generally higher than IOP 13b. Note that these higher $\alpha$ and $\gamma$ values are associated with a thicker fog layer with CTH values from 45 m to 75 m compared with the maximum height of 50 m for IOP 13b. It is 505 worth noting that one hour later, around 2300 UTC, the LWC # 4 profile increases slightly with height, although the transition occurs one hour later, but in line with the transition period defined by Dione et al. (2023), which began at 2300 UTC. The associated temperature profile is still globally stable, but some warming is observed between 65 and 120 m height, which could be explained by the change in wind direction. Below this, the layer between 25 and 65 m has begun to destabilize, possibly reflecting the onset of vertical motions. During the transition (profile # 5) the entire layer was much colder and 510 unstable. The LWC exhibited a two-layer profile with an adiabatic tendency up to 50 m height and lower values in the upper part.

Surprisingly, reverse LWC trends were observed for profiles # 6 and # 7 with $\alpha$ <0 even though the thin-to-thick transition had already occurred one hour earlier. In addition, LWC values at the ground reached 0.25 g.m$^{-3}$ which are the highest values measured on all IOPs. Note that consistently, high radar reflectivities were also observed near the ground in Fig. 10c. The 515 corresponding lapse rates are fairly constant through the layer constrating with the stable profiles observed during the thin phase. The LWC profile becomes increasing again with a slightly positive $\alpha$ at profile # 8 sampled when the fog is 240 m thick, and also in stratus where $\alpha$ reaches 0.56 with a vertical profile of temperature that has finally destabilized with $\gamma = 1.82$. The following vertical profiles measured inside the stratus reveal the same microphysical and thermodynamical properties (not shown) as those observed in IOP 14.





**Figure 10.** Same legend as Fig. 9 but for IOP 11





Consequently, it is clear from Fig. 10f that the temporal evolution of $\alpha$ and $\gamma$ during the fog life cycle follows the same trend as for IOP 14, from negative values when the fog is optically thin to positive values after the transition to thick fog, except for the sharp decrease in $\alpha$ for profiles # 6 and # 7, which disrupts the overall trend.

### 4.2.3   IOP 6 case study

IOP 6 took place during the night of January 5-6, 2020 and was characterized by anticyclonic conditions over France, associated
with clear sky and negligible wind. Surface radiative cooling became established on the "Supersite" after sunset, and as the south-westerly wind began to blow, ground visibility suddenly dropped by more than 10 km in 10 minutes (Fig. 11a). This sudden fog formation at 2037 UTC is associated with the appearance of an upper liquid water layer on the BASTA reflectivity (Fig. 11c), which was also detected at different heights of the towers surrounding the "Supersite" (not shown). Fog formation was spread out over time at the various instrumented sites and was observed earlier and later for the sites located to the
southwest and northeast of the domain, respectively (not shown). This suggests that a mesoscale advection coming from the west with moist maritime characteristics favoured the formation of fog in addition to radiative cooling near the ground. Therefore, we rather classify this episode as a radiative-advective fog, although Dione et al. (2023) estimated that the formation was mainly driven by radiative cooling at the surface.

As discussed in section 2.3, the transition from thin-to-thick fog occurred very rapidly (50 minutes) after fog formation
based on LW flux measurements. However, the vertical developpment of the fog layer was temporarily stopped when the wind direction shifted from southwest to southeast, and it took around 2h30 for the fog to deepen again around midnight. Consequently, the other parameters indicate a transition occurring much later between 0000 for TKE and 0230 UTC for LWP, in agreement with Dione et al. (2023), who estimated that the period lasted 2 hours from midnight. Finally, the fog dissipated completely at 0928 UTC, around 1h30 after sunrise, as it lifted into a deep stratus that persisted throughout the day until it
lowered and formed fog again around 1600 UTC.

Among the twenty-seven vertical profiles validated during IOP 6, nine representative profiles selected between 2152 and 0300 UTC are reported in Fig. 11 to illustrate the evolution of the microphysical and thermodynamical properties during the fog development period. In addition, the $\alpha$ and $\gamma$ values for the last four profiles when the fog remained adiabatic, despite the decrease in thickness of around 100 m and the rise in stratus (profiles # 10 to # 13 from 0600 to 1100 UTC), are also indicated
in Fig. 11f.

As a result, they all show quasi-adiabatic characteristics with positive values of $\alpha$ and $\gamma$, except during the period where developement is stopped from 2230 to 0000 UTC. This confirms that IOP 6 became optically thick very early as indicated by LW measurements. Note that for this case, Dione et al. (2023) classified the period from 2040 to 0000 UTC as stable, considering that the transition period started at midnight, as they observed that $\alpha_{eq}^{closure}$ values remained negative with MWR LWP values
$< 10$ g.m$^{-2}$. In contrast, the first three profiles in Fig. 11d-e show a clear increase in LWC with height associated with the quasi-adiabatic lapse rate. This suggests that other thresholds characterizing the transition with remote sensing measurements should be defined to better detect the transition when the fog life cycle is perturbed by non-local processes.





**Figure 11.** Same legend as Fig. 9 but for IOP 6. Visibility in (a) is issued from the Jachère site.





During the period corresponding to the wind rotation, the CTH dropped from 80 to 30 m in height and a large variability appears on the LWC vertical profiles. The $\alpha$ values decrease and become negative although the vertical profiles do not reveal decreasing values with maximum values near the ground, but rather a constant LWC with height as seen on profile # 4, except for the minimum values of $\alpha$ = -1.38 when the fog is at its thinnest (profile # 5). During this period, the lapse rate decreased and remained almost neutral ($\gamma \approx 0$) but did not reach low negative values as in previous cases. It is likely that the layer could not return to thermally stable conditions because the time period is too short to allow coupling with the surface, despite the ground temperature lost 1 °C. From midnight, fog thickness increased again. In the same way, the LWC increased with height, and the $\alpha$ values are positive and range between 0.51 and 1.35 until the end of the episode, including in the stratus that followed dissipation (Fig. 11g, dotted blue line). As a result, the corresponding temperature profiles are very close to the moist adiabatic lapse rate with $\gamma$ between 0.71 and 0.94.

This event therefore corroborates the fact that the profiles sampled after the thin-to-thick transition exhibit quasi-adiabatic characteristics with positive values of both $\alpha$ and $\gamma$, except when dynamic conditions interrupt vertical developpement and lead to a significant reduction in fog thickness.

## 4.3 Correlation between the microphysical and thermodynamical properties

The four case studies presented above have revealed a similar temporal evolution of the vertical profiles of microphysical and thermodynamical properties with contrasting patterns before and after the transition from optically thin-to-thick fog. Values of $\alpha$ and $\gamma$ of the 140 validated profiles of the campaign are reported on Fig. 12a with empty, full circles, and diamonds, for thin and thick fogs and stratus, respectively, to highlight the contrast between the different phases.

A large majority of profiles performed during the thin phase are located in the lower left part of the graph, corresponding to negative $\gamma$ and $\alpha$ values, while those performed in thick fog are located in the upper right part corresponding to positive or near-zero $\gamma$ and $\alpha$ values. This trend is particularly marked for the lapse rate. For instance, $\gamma < -0.5$ for all the profiles performed in thin fog except the #3 profile of IOP 14. Similarly, $\gamma > -0.5$ for all the profiles in thick fog except the two specific reverse LWC profiles of IOP 11. LWC profiles are a bit more scattered but $\alpha < 0$ for 74 % of the thin cases and $\alpha > 0$ for 75 % of the thick ones. The lowest $\alpha$ values, between -5 and -8, correspond to very stable conditions with $\gamma < -10$. These profiles performed during IOP 2b are associated with the lowest CTH values < 10 m (Fig. 12c) and also with very low LWP values < 2 g.m$^{-2}$ (Fig. 12b). In contrast, optically thick fogs and stratus by fog lifting are associated with the highest values of LWP and CTH, as illustrated previously by IOP 6c, 13b and 14. In between, intermediate samples of $\gamma$ ranging between -10 and 0 are associated with intermediate values of $\alpha$ around 0. These profiles are associated with CTH values reaching 50 m but not necessarily with significantly higher LWP values.

An interesting difference when comparing the thin phase of these fogs concerns the amplitude of the adiabatic and lapse rate fractions. For fogs that remained optically thin during their life cycle, $\gamma$ ranges from -30 to -15 on average and $\alpha$ from -4 to 0. Whereas for those that underwent a transition to thick fog, $\gamma$ ranges from -10 to 0 and $\alpha$ fluctuates around -1.5/-1. This would suggest that fogs that remained thin are associated with much stable conditions and stronger reverse LWC profiles. But there are only 3 thick cases sampled before the transition and more data are needed to confirm this result.



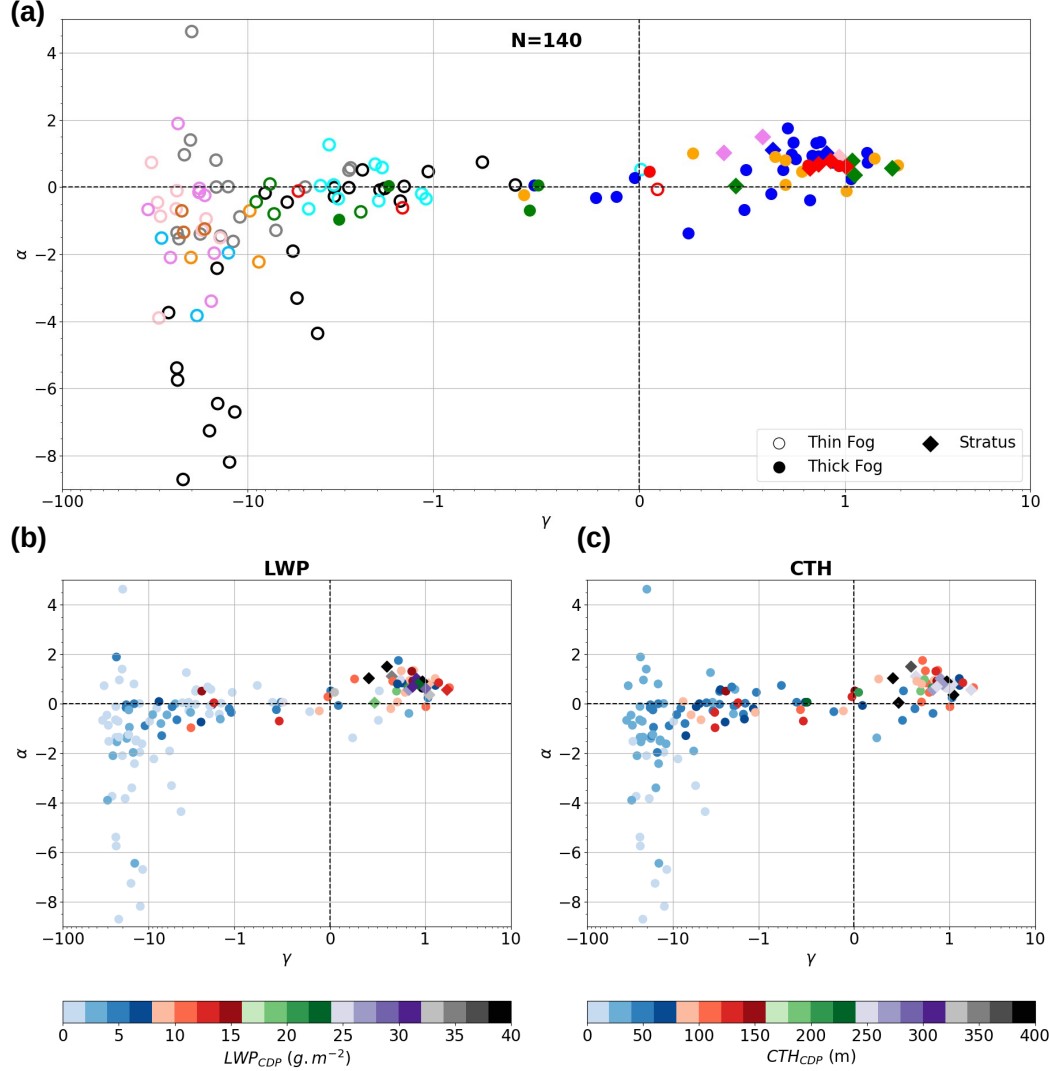

**Figure 12.** Adiabatic fraction as a function of the lapse rate fraction in logarithmic scale, over the 12 IOPs : (a) colored with IOPs as in Fig. 4, thin and thick fog are indicated by empty dots and filled dots, respectively, (b) colored with LWP values issued from the CDP measurements, (c) colored with CTH values issued from the CDP measurements. Fog and stratus are indicated by dots and diamonds respectively.

In conclusion, the analysis of the 140 profiles sampled during the experiment reveals significant differences between thin and thick fogs, with adiabaticity and lapse rate fractions being much lower for thin fogs ($\alpha$<0, $\gamma$<-0.5) than for the thick ones and stratus ($\alpha$>0, $\gamma$>-0.5). This confirms the temporal evolution from reverse LWC profile and stable conditions during the thin
phase to quasi-adiabatic features after the transition as depicted by the 3 deepest IOPs.





## 5   Discussion

The CDP data collected below the tethered balloon have revealed reverse trend of LWC profile (LWC being maximal at the ground and decreasing with altitude) when stable temperature conditions exist during the optically thin phase of fogs. Oliver et al. (1978) obtained similar shape of LWC profile in advective-radiative fog by using a second-order closure model of

turbulence. But in their simulation, the fog forms in an unstable mixed layer initiated by the advection of a stable boundary layer over a region with warmer surface temperature. Toledo et al. (2021) mentionned that such shape should exist to explain their observations of $\alpha_{eq}^{closure} < 0$. By using cloud radar reflectivity measurements Wærsted et al. (2017) also retrieved higher LWC values near the ground for a thin fog event. But to our knowledge, only two cases of LWC vertical profile decreasing with height have been reported in the litterature. Okita (1962) measured one profile in a very thin layer $< 20$ m just after the fog

formation. More recently, one of the 2 profiles through the radiative fog sampled during IOP1 of the LANFEX experiment in Boutle et al. (2018) presents also higher values at surface, with CTH $< 40$ m and LWP $< 15$ g.m$^{-2}$. Measurements presented here then suggest that such features are ubiquitous in optically thin fogs. Because such cases correspond also to geometrically thin fog layer (typically from 10 to 50 m height), the LWP is very low (typically a few g.m$^{-2}$) and these properties can hardly be captured by current remote sensing instruments (first available gate of the cloud radar from 25 to 37.5 m height).

We therefore use these in situ data to compute a local adiabaticity $\alpha$ and a lapse rate fraction $\gamma$ using linear regressions in order to best fit the vertical profiles of LWC and temperature, respectively. This method allows to remove the impact of the mixing at fog top and it is not very sensitive to significant gradients between two consecutive altitude ranges. As a result, it provides valuable information to investigate departure from the adiabatic model and to document the evolution during the fog life cycle. However, these calculations must be interpreted with great caution when fog layer is thin or very thin (lower than

10 m), because it may lead to very low or very high values of $\alpha$ and $\gamma$ when LWC and temperature gradients are calculated on a reduced number of data. Figure C1a in Appendix illustrates that the methods work perfectly for a sharp decrease in LWC at an altitude of 10 m, resulting in a very low value of $\alpha$ at -8.31. But Fig. C1c-d report measurements from the first profile sampled during IOP 2a, which exhibit the highest $\alpha$ value (4.63) with one of the lowest $\gamma$ values (-20). This particular profile was performed at the very beginning of fog formation, when condensation first occurred at a height of around 20 m before

the surface, thus distorting the slope calculation and giving a misleading result. This is also the case for profile # 4 of IOP 13, where slightly high values of LWC at 30 m height combined with some low values at 10 m height lead to an $\alpha$ value of 1.4, whereas the profile appears to be fairly constant with altitude. Thick fogs are also subject to crude approximation when profiles are not monotonic, as LWC profiles # 5 of IOP 11 and IOP 14. In such cases $\alpha$ values obviously can not represent the shape of the profiles.

Nevertheless, despite such uncertainty, this analysis has shown that these reverse LWC profiles sampled during the thin phase evolve towards quasi-adiabatic features with increasing LWC values with altitude and a neutral to slightly unstable temperature lapse rate, when the transition to optically thick fog occurs. The 3 deepest episodes presented previously provide convincing elements that highlight such evolution of the vertical profiles during the fog life cycle. The number of sampled profiles, however, is limited, and these fog life cycles were perturbed by non-local processes such as low clouds advection or





changes in wind direction , which complicate the interpretation of the observations. On the whole, however, the 140 profiles
sampled during the experiment provide a robust assessment of this finding.

To determine the transition time independently of the tethered balloon measurements, we defined 5 thresholds for LW flux,
TKE, vertical temperature gradient, CTH and LWP. We found that the threshold value of 30 $g.m^{-2}$ for LWP proposed by
Wærsted et al. (2017) is too high for our cases and that a values of 15 $g.m^{-2}$ is more suited. Dione et al. (2023) also found

that the adiabaticity from closure $\alpha_{eq}^{closure}$ exceeds 0.5 when LWP > 20 $g.m^{-2}$. In fact, in general, the fogs sampled at the
Charbonnière site during SOFOG3D have lower LWP values than previous studies at SIRTA (Wærsted et al., 2017; Toledo
et al., 2021), but similar to IOP1 of LANFEX (Boutle et al., 2018). Note that LWP measurements at Agen in the Garonne
valley during SOFOG3D were much larger (Fathalli et al., 2024) (in prep), confirming that site characteristics determine fog
layer properties. Note also that if the 30 $g.m^{-2}$ LWP threshold is too high, the CTH threshold is adequate. Consequently, this

suggests that the transition depends more on the geometric extent of the layer than on the LWC amount.

Transition time estimates from these different parameters are in agreement during IOPs 11 and 14, during which there was a
clear thin-to-thick transition (uncertainty periods of 0h25, and 0h48, respectively), but they are more dispersed during IOPs 9a
and 6c (uncertainty periods of 2h29 and 3h41, respectively) since the fog became optically thick immediately after its formation
or shortly after, respectively. It is worth noting that the evolution of the LWC profiles is consistent with these transition time

estimations, with reverse profiles before the transition evolving towards an increase with altitude after the transition. With
respect to IOP 6c, the LWC profiles provide a finer scale understanding of the fog evolution confirming that the layer is already
thick very shortly after the fog onset and becoming temporarily thin as the wind direction changes. Recently, Dione et al.
(2023) proposed to additionally use $\alpha_{eq}^{closure}$ to better discriminate the different phases of the fog life cycle (stable, transition,
adiabatic and dissipation). The resulting durations of the transition from stable to adiabatic phase for IOP 11, 14 and 6 are 3h30,

1h30 and 2h, respectively (their Table 2). These deviations from the uncertainty periods derived above mainly arise from their
criteria that determines the end of the transition phase, namely when $\alpha_{eq}^{closure}$ > 0.5. Indeed, the time series reported in Dione
et al. (2023) indicate that the evolution of $\alpha_{eq}^{closure}$ satisfactorily reproduces periods of thin and thick fog, with promising strong
variations during the fog life cycle. Further case studies are needed to better define the key parameter values for characterizing
the transition.

We have shown, however, that some uncertainty in the calculation of $\alpha_{eq}^{closure}$ is introduced by the $LWC_0$ value derived
from the visibility by the parameterization of Gultepe et al. (2006), which does not take into account the actual droplet number
concentration. CDP measurements provide accurate measurements of $LWC_0$. Figure 13 reveals that $LWC_0$ values are higher
as $\alpha$ decreases, although there is a large dispersion, and a significant reduction of $LWC_0$ values for $\alpha > 0$. This suggests that
LWC near the ground decreases after the transition, as the LWC profile is changing from reverse to increasing with height. This

is consistent with observations of Mazoyer et al. (2022) that reveal a decrease of the LWC measured near the ground during
the mature phase of fogs, resulting from the evaporation due to surface warming induced by infrared radiation emitted by the
fog itself.

Profiles sampled after the transition exhibit continuous increase of the LWC with height but with a large variability of the
adiabaticity, profiles # 6, # 7 and # 8 of IOP 6 are very close to the adiabatic, whereas profiles # 4 of IOP 14 and # 8 of IOP11



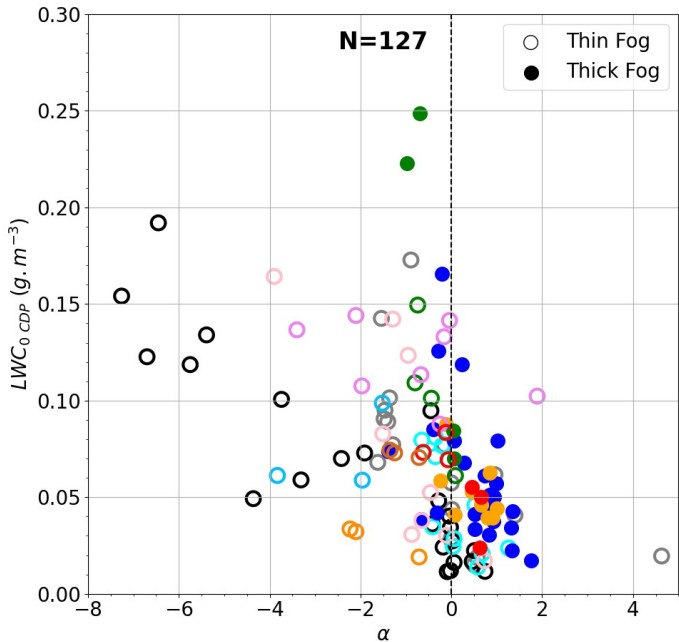

**Figure 13.** $LWC_0$ as a function of the local adiabaticity $\alpha$ for the 12 IOPs with the same symbols as Fig. 4 and 12.

are much more diluted throughout the fog layer. We have seen that $\alpha_{eqCDP}^{closure}$ values for the few thick profiles with CTH > 150 m sampled with the CDP fluctuate around 0.4, which is below the convergence value $\approx 0.6$ reported for fog conditions in Toledo et al. (2021) and Dione et al. (2023). During SOFOG3D, 13 profiles were performed in stratus cloud. They reveal that the adiabaticity is also higly variable, with profile # 9 of IOP 13b adiabatic and other very diluted such as # 8 of IOP 14 or # 9 of IOP 11. The mean $\alpha_{eqCDP}^{closure}$ value reaches 0.61 with 1st and 3rd quartiles of 0.48 and 0.67, respectively, and the mean $\alpha$

value reaches 0.76 with 1st and 3rd quartiles of 0.56 and 1.01, respectively. These statistics are in agreement with the typical adiabaticity of 0.7 observed in stratocumulus clouds from airborne measurements (Brenguier et al., 2011; Cermak and Bendix, 2011; Braun et al., 2018). This confirms that these lower adiabaticity values derived from CDP measurements result from the actual properties of the sampled fogs. Other case studies are obviously necessary to assess whether this is a general trend and to better understand the processes that explain such characteristics.

As pointed out previously, the 3 LWC profiles sampled after the transition during IOP 11 depict very unexpected trend. Profile # 5 presents a two-layer structure with adiabatic increase until 50 m height and reduced values above. Such a shape has already been observed in a fog formed by stratus lowering (Fathalli et al., 2022) but it remains difficult to explain. Surprisingly reverse LWC trends are observed for profiles # 6 and # 7 sampled less than one hour later with the highest $LWC_0$ values measured near the ground (0.25 g.m$^{-3}$). Interestingly, these 2 $LWC_0$ values come out of all the samples in Fig. 6, Fig. 11a

and Fig. 13. Given that, these profiles were sampled 1h30 after the transition when the fog layer was well developed, with



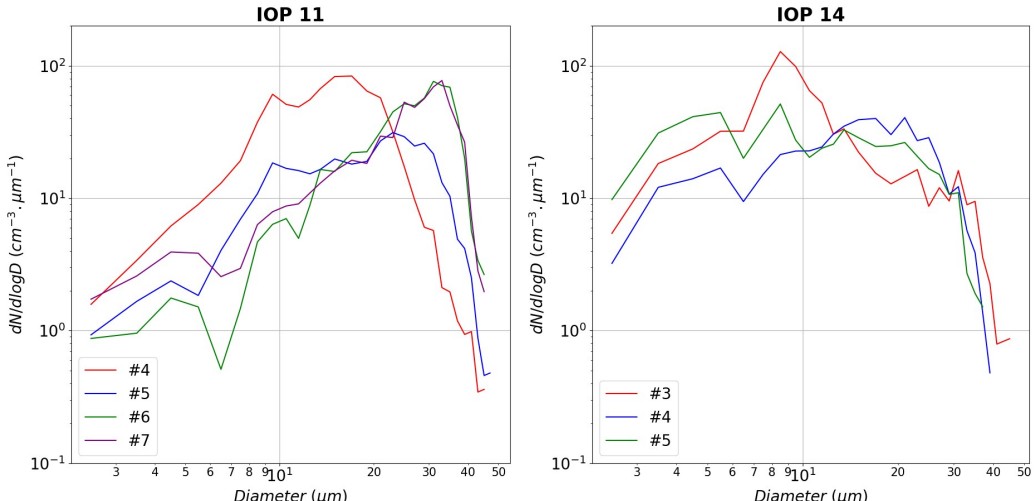

**Figure 14.** Droplet number size distributions measured by CDP in the 15 m thick layer above the surface for profiles (a) # 4 to # 7 from IOP 11, and (b) # 3 to # 5 from IOP 14.

LWP > 10 g.m$^{-2}$ et CTH > 120 m, and the lapse rate slightly unstable: it is very unlikely that these reverse profiles result from the condensation at surface due to radiative cooling as those observed during the thin phase of radiation fogs.

Indeed, these high values of $LWC_0$ result from the existence of a mode of large droplets as attested by Fig. 14a, which shows the evolution of the droplet number size distributions (DSD) measured by the CDP at the surface during IOP 11. Initially centered around 15-20 µm (profile # 4 red line), the DSD broadens and shifts towards larger sizes during the transition (profile # 5 blue line). Finally, for profiles # 6 and # 7, the DSD near the ground are very similar with a mode as large as 35 µm. In contrast, for IOP 14 (Fig. 14b), the DSD before the transition exhibits a mode below 10 µm, which is further shifted to about 20 µm. Note that droplets as large as 30 µm also exist but at a much lower concentration. DSD with such mode of large droplets have been observed in many experiments as reported in Wendish et al. (1998); Gultepe et al. (2009); Niu et al. (2011); Formenti et al. (2019); Mazoyer et al. (2022); Wagh et al. (2023). Usually, they are mainly oberved in bimodal DSD and are attributed to a mass transfer from the smaller droplets into the larger ones through collision–coalescence or Ostwald ripening processes, and droplet removal through sedimentation. It is remarkable here that the DSD # 6 and # 7 of IOP 11 have a single mode and are very narrow with high concentration of large droplets. In addition, vertical profiles reveal that LWC has decreased in the upper part of the fog layer, contributing also to the reversal of the LWC trend with height. It is therefore likely that collision–coalescence and sedimentation processes actively contributed to produce such feature by redistributing the liquid water through the fog layer from the top to the ground. A more detailed analysis of the evolution of the vertical profile of the microphysical properties is currently conducted to confirm this hypothesis and will be presented in a companion article to be submitted soon.



## 6  Summary and conclusions

The aim of this study was to document vertical profiles of microphysical and thermodynamic properties in fog layers from in situ measurements collected under a tethered balloon during the SOFOG3D field campaign. Droplet size distributions recorded by an adapted CDP provide 140 vertical profiles of LWC, which allow an exhaustive analysis of 8 thin fogs (thickness < 50 m) and 4 developed layers, including 3 episodes exceeding 200 m in thickness. These data reveal that reverse trend of LWC profile (LWC being maximal at the ground and decreasing with altitude) are ubiquitous in optically thin fogs, while quasi-adiabatic

features with LWC values increasing with altitude are mainly observed in well-mixed optically thick fogs. We used remote sensing instruments (microwave radiometer and Doppler cloud radar) and surface measurements to determine the thin-to-thick transition time, and defined 5 thresholds for LW flux, TKE, vertical temperature gradient, fog top height and LWP, enabling us to assess an associated period of uncertainty. We found that a LWP threshold value of 15 $g.m^{-2}$ is more suited for the 4 thick fogs sampled at the Charbonnière site. This analysis also suggests that the transition depends more on the geometric extent of

the layer than on the LWC amount.

The CDP data are used to compute the equivalent fog adiabaticity from closure and compare to the value defined by Toledo et al. (2021) which is derived from remote sensing instruments, 2-m height visibility, and an one-column conceptual model of adiabatic continental fog assuming that LWC linearly increases with height. The comparison shows a large variability that results mainly from the parameterization used to estimate LWC at ground from the visibility, in the equivalent fog adiabaticity

from closure, but their evolution as a function of the fog thickness follows the same trend, demonstrating the consistency between both methods. We found larger negative values for thin layers, associated to low LWP values, that represent a specificity of the fog layers sampled at the Charbonnière site during the SOFOG3D experiment. In situ data are further used to investigate the actual fog adiabaticity and lapse rate fraction by using linear regressions to best fit the vertical profiles of LWC and temperature, respectively. We presented an analysis of 4 episodes that highligths that reverse LWC profiles, when stable temperature

conditions exist during the optically thin phase of fogs, evolve towards quasi-adiabatic features with increasing LWC values with altitude, and neutral to slightly unstable temperature lapse rate, when fog becomes thick. This study reveals that non-local processes, such as low cloud advection or changes in wind direction, perturb the fog life cyle and modulate the thin-to-thick transition. On the whole, however, the 140 profiles sampled during the experiment provide a robust assessment of the vertical profile temporal evolution. We also found that LWC at ground is higher during the thin phase, and significantly decreases as

the profile is changing from reverse to increasing with height. But this trend could be balanced when collision-coalescence and sedimentation processes redistribute the LWC through the fog layer from the top to the ground.

This study provides new insights into the evolution of LWC profiles during the fog life cycle that would help constrain numerical simulations. Although this analysis is based on 140 vertical profiles, the number of samples collected during the thin-to-thick transition remains limited. More continuous observations are therefore required to examine the triggering factors

of this transition and to disentangle the local and non-local processes involved in the fog life cycle. The method developed to evaluate the local adiabaticity by fitting the vertical profiles of LWC allows to remove the impact of mixing at the fog top.





However, the results revealed large variability among the differents cases, with some profiles highly diluted. More analysis is then needed to explain such departure from the adiabatic model in well-mixed fogs.

To take a step forward, we are currently conducting a detailed analysis of the vertical profiles of droplet size distribution measurements. Following Mazoyer et al. (2022), the main objective is to investigate the vertical and temporal evolution of the droplet size as a function of the number concentration. The vertical profile dataset is extended with constant-height sections at various altitudes within fog layers, which were performed for TKE measurements. Reflectivity and Doppler velocity along the sight of water drops, measured by the BASTA cloud radar, also provide valuable information on the dynamical and microphysical properties of the fog layer (Vishwakarma et al., 2023; Dione et al., 2023). A scanning unit was deployed around 1 $km$ away from the vertical pointing unit of the Charbonière site, which allowed for the first time a volume sampling of a fog layer. Investigation of these data in synergy with CDP measurements would improve the three-dimensional retrieval of microphysical properties and provide new insights to better understand the key processes driving the microphysical evolution during the fog life cycle.

## Appendix A: Determination of the thin-to-thick transition time

### A1 IOP 11

As for IOP 14 (Fig. 2), there is a consistency in the temporal determination of all the thresholds (Fig. A1), which range from 0030 UTC (TKE, green line) to 0055 UTC (vertical temperature gradient, blue line), i.e. an amplitude of 25 minutes, (Table 2).

### A2 IOP 6

A significant variability is observed in the temporal determination between all the thresholds (Fig. A2). Downward longwave radiation strongly increases at the fog formation, leading to a decrease in $LW_N$, reaching 5 W.m$^{-2}$ at 2126 UTC, 49 minutes after the fog formation at 2037 UTC (Fig. A2b). At this point, a decrease in vertical temperature gradient is observed, with the fog layer closest to the ground becoming almost neutral and therefore almost reaching the vertical temperature gradient threshold (Fig. A2d). An increase in LWP and CTH is also observed at this time, with the latter close to the 110 m threshold (Fig. A2f). Then, all the variables diverge from their respective threshold, with the exception of $LW_N$. When the fog deepens vertically at around 0000 UTC (Fig. 11), TKE exceeds its threshold, while CTH, vertical temperature gradient and LWP thresholds are reached later, between 0101 and 0136 UTC.

## Appendix B: Fog equivalent adiabaticity from closure

Figure B1 illustrates the comparison between the values of $\alpha_{eq}^{closure}$ and LWP as well as CTH, using the measurements retrieved from the HATPRO radiometer and the BASTA radar respectively. The results are quite similar to that using the CDP measurements (Fig. 6). However, as in Fig. 6, three vertical profiles exhibit superadiabatic behaviour. The two profiles during POI 2a (in grey) show $\alpha_{eq}^{closure}$ values greater than 1 due to low $LWC_0$ values, which are associated with low LWP and CTH





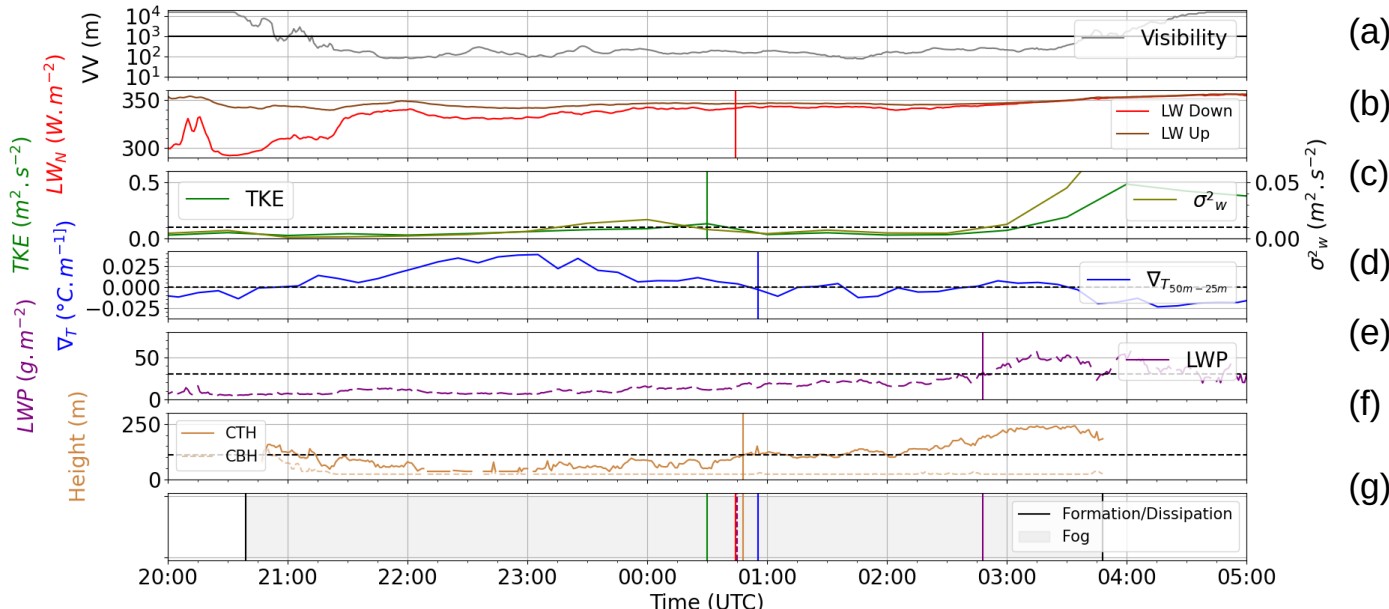

**Figure A1.** Same legend as Fig. 2 but for IOP 11.

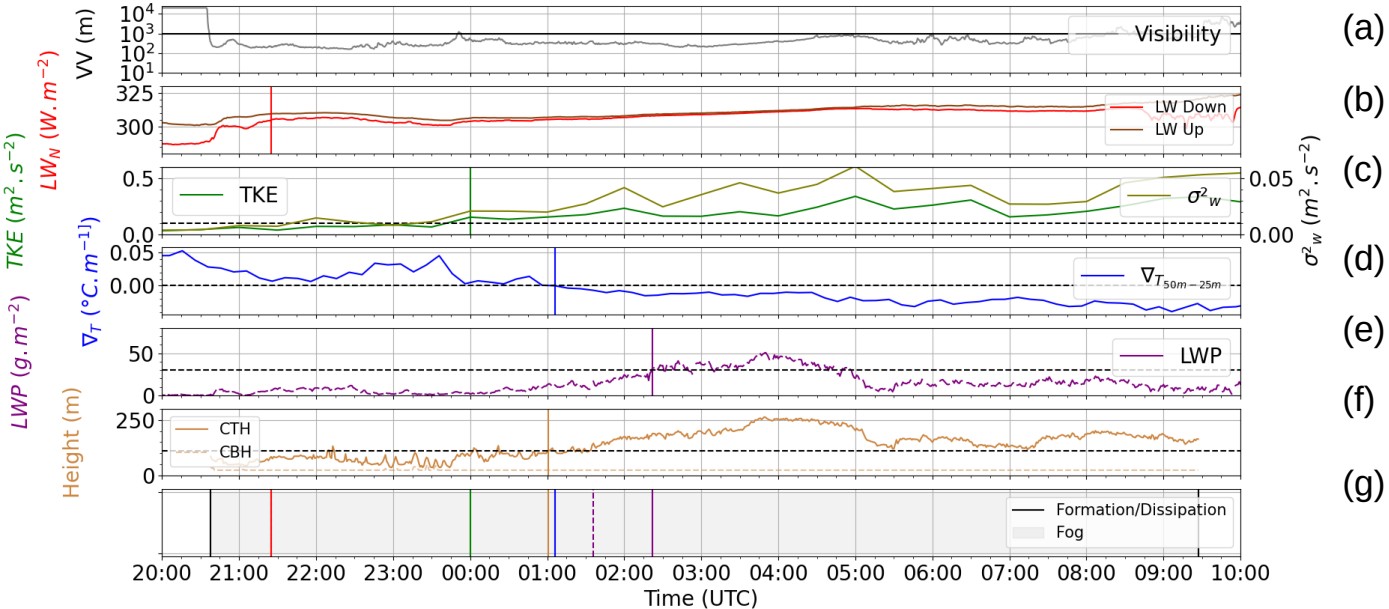

**Figure A2.** Same legend as Fig. A1 but for IOP 6



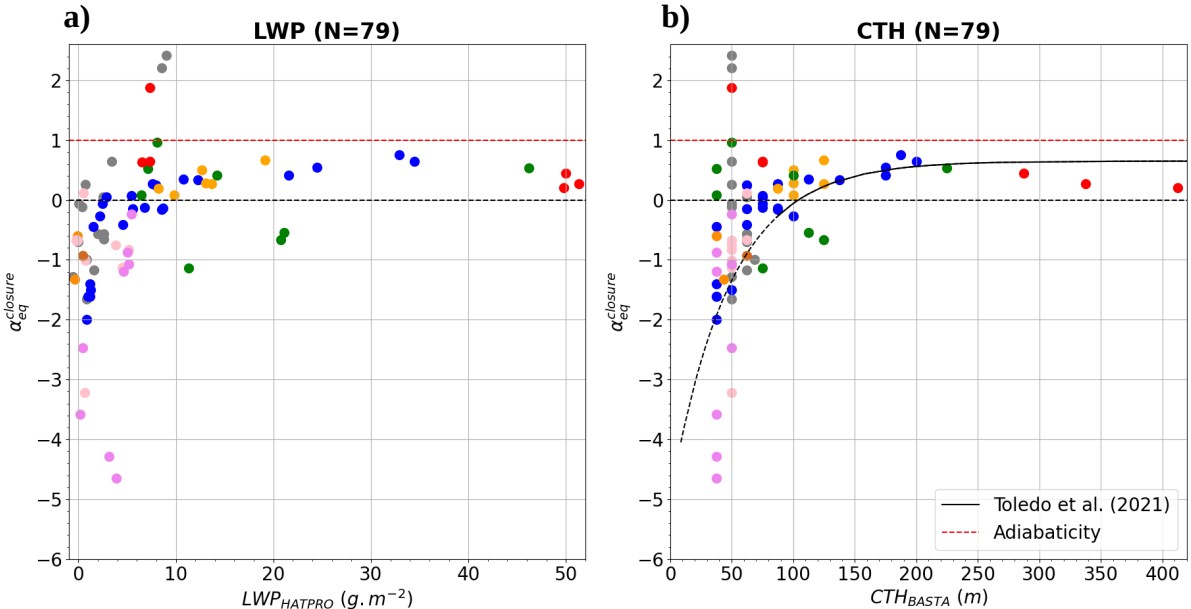

**Figure B1.** Same legend as Fig. 6 but using LWP and CTH derived from the HATPRO radiometer and BASTA radar, respectively.

values as well, result in a strong overestimation of $\alpha_{eq}^{closure}$. In addition, the superadiabatic vertical profile during IOP 14 (in red) is associated with a value of CTH just above the detection limit of the BASTA radar.

## Appendix C: Vertical microphysical properties

Figure C1a,b shows the vertical profiles of temperature and LWC during IOP 2b. This IOP is characterized by three distinct fog periods, between 1924 UTC and 2110 UTC, associated with a 130 m thick fog, between 0002 UTC and 0136 with a 60 m vertically thick fog and between 0331 UTC and 0513 UTC, associated with a thin fog with a maximum CTH of 30 m. These dissipations may be explained by the presence of temporary medium and low clouds detected by the BASTA radar at around 2130 UTC and 0100 UTC, close to 3000 m and 1100 m, respectively. The two vertical profiles illustrated in Fig. C1a and C1b

are representative of the last fog episode, characterized by very low values of $\alpha$ and $\gamma$. Figure C1c,d shows the vertical profiles of temperature and LWC for the first sample during IOP 2a, which reveals a sharp increase in LWC below the fog top, related to condensation occurring at 20 m A.G.L before the surface.

*Data availability.*   All the data used in this study are hosted by the French National Center for Atmospheric Data and Services (AERIS) at the following links: https://doi.org/10.25326/XX. The CDP data files are being finalized for submission to the database. https://doi.org/10.25326/89

(Canut, 2020), https://doi.org/10.25326/135 (Delanoë, 2020) and https://doi.org/10.25326/207 (Martinet, 2021). Data access can be provided by following the conditions fixed by the SOFOG3D project.



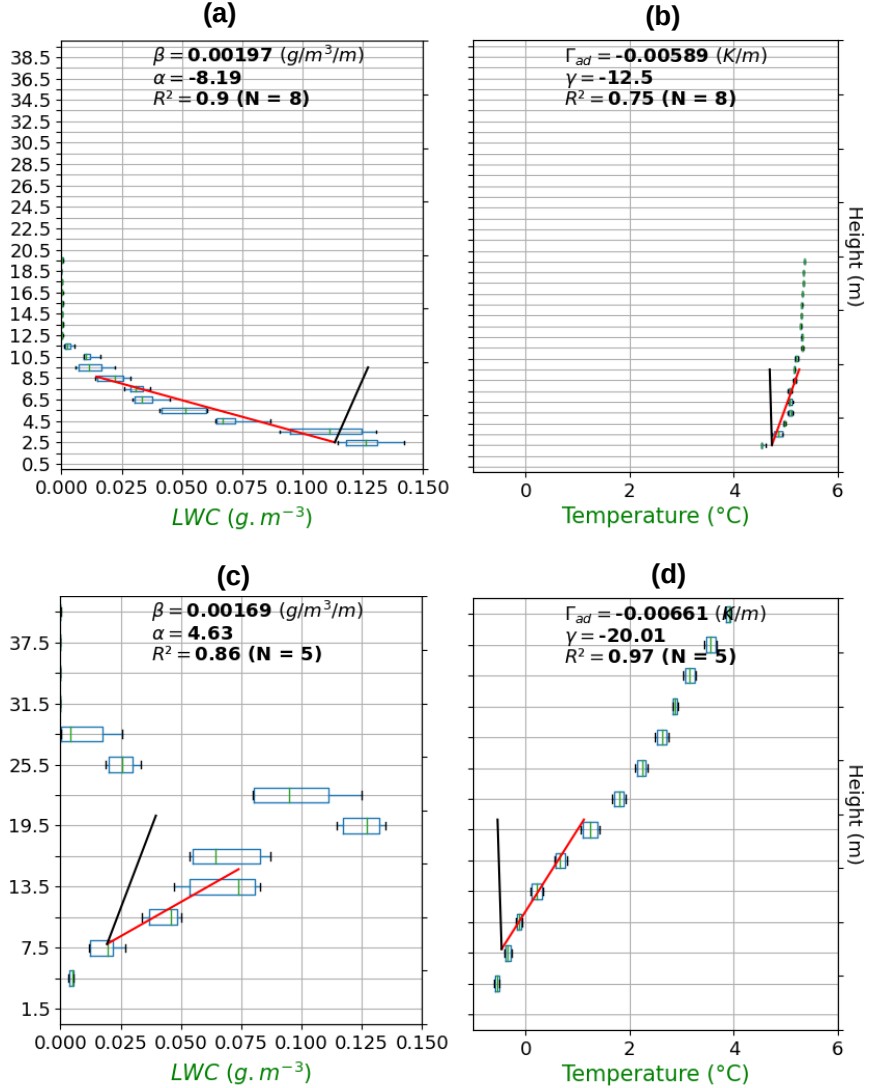

**Figure C1.** Vertical profile of (left) LWC (right) temperature during (a,b) the ascent between 0336 UTC and 0339 UTC of IOP 2b (c,d) the ascent between 2139 UTC and 2141 UTC of IOP 2a. Are indicated (left) the theoretical adiabaticity, the adiabaticity and the coefficient of determination of the LWC linear regression with the number of points used in brackets (right) the moist adiabatic lapse rate, the lapse rate and the coefficient of determination of the temperature linear regression with the number of points used in brackets.

*Author contributions.* TC processed in-situ data and performed the analysis supervised by FB and CL; Conceptualization, methodology, investigation, validation, data curation, formal analysis, visualization, writing—original draft. FB was the PI of the SOFOG3D project, designed the field campaign and led the tethered balloon operations; Project administration, resources, conceptualisation, methodology, investigation, validation, formal analysis, supervision, writing—original draft. CL was the PI of the high resolution modeling task and co-led the processes task, Conceptualization; methodology; supervision; writing—original draft. PM was the PI of the microwave radiometer



network; Supervision, resources, investigation, data curation. JD was the PI of the cloud radars deployment; Resources, supervision. SJ : data curation. MF : investigation, data curation.

*Competing interests.* The contact author has declared that none of the authors has any competing interests.


*Acknowledgements.* The SOFOG3D field campaign was supported by Météo-France and ANR through grant AAPG 2018-CE01-0004. Data are managed by the French National Center for Atmospheric Data and Services (AERIS). The MWR deployment at Charbonière site was carried out thanks to support by the Köln University. MWR data have been made available, quality controlled and processed in the frame of CPEX-LAB (Cloud and Precipitation Exploration LABoratory, www.cpex-lab.de), a competence center within the Geoverbund ABC/J by acting 750 support of Ulrich Löhnert, Rainer Haseneder-Lind and Arthur Kremer from the University of Cologne. This collaboration is driven by the European COST actions ES1303 TOPROF and CA18235 PROBE.




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
