# Peer review of "Vertical Profiles of Liquid Water Content in fog layers during the SOFOG3D experiment"

_EGUsphere, 2024_

## Referee Comment (RC2)

**Review of "Vertical Profiles of Liquid Water Content in fog layers during the SOFOG3D experiment" by Costabloz et al. submitted to Atmospheric Physics and Chemistry**

This paper studies vertical profiles of liquid water content (LWC) observed in fog layers during a field campaign carried out in southwestern France. The twelve fog events sampled during the campaign concern "thin" fogs occurring in stable conditions, characterized by decreasing LWC and increasing temperature as a function of altitude and vertically developed (or "thick") adiabatic fog, characterized by increasing LWC and decreasing temperature as a function of altitude.

The work is original in that it relies on a comprehensive set of in-situ measurements of fog microphysics, complemented by remote sensing using a cloud radar and microwave radiometer.

The paper includes several interesting investigations that make it worthwhile to consider this paper for publication in Atmospheric Physics and Chemistry: Vertical profiles of LWC (measured by a cloud droplet probe, CDP) and temperature are analysed at different stages of the fog life cycle. The conditions that lead to thin-to-thick transition are investigated for four fog events using remote sensing measurements, surface measurements and in-situ vertical profiles. This is important to evaluate how the representation of transition can be improved in model simulations. Further, this analysis is compared to a conceptual model of fog adiabaticity proposed and exploited earlier in the literature (Toledo et al. 2021 and Dione et al. 2023) and provides a detailed assessment of the performance of this conceptual model.

The paper is clearly written and well organized. However, two topics require thorough attention and revisions, while others are more minor.

**Major comment 1: derivation and uncertainty of fog adiabaticity.**

Fog adiabaticity is studied extensively in the paper, both from in-situ LWC measurements and using the conceptual model proposed by Toledo et (2021). The paper presents several methods to derive fog adiabaticity.

1. Equivalent adiabaticity from Closure (Eq. 5), where input variables are derived from remote sensing measurements and horizontal visibility
2. Equivalent adiabaticity from Closure (Eq. 5), where input variables are derived from in-situ CDP measurements
3. Adiabaticity from fitting LWC profiles from the ground to just below the diluted layer.

**Section 3.2 and 3.3** are dedicated to comparing the first two definitions,

The balloon ascent (that carries the CDP sonde) takes 15-40 minutes to cover the vertical extent of the fog layer, depending on its depth. In the equivalent adiabaticity formulation (Eq. 5), there is an inherent hypothesis that the $LWC_0$ measured at the surface is consistent with the integral of LWC (the liquid water path, LWP) and the fog layer depth (provided by CTH). We know that CTH, LWP and $LWC\_0$ can vary significantly in 20-30 minutes.

Q1.1 Hence what is the uncertainty in deriving equivalent adiabaticity (definition 2) from CDP measurements given the temporal variations of LWC during the ascent or descent of the balloon?

Q1.2 What temporal smoothing or averaging is used for CTH, LWP and visibility to derive equivalent adiabaticity (definition 1).

Q1.3 What is the impact of these uncertainties on the comparisons of the two alpha-closures made in figure 4?

Q1.4 Line 310-311: the alpha-closure derived for IOP11 (22:09 to 22:23) is 0.53, while the same alpha-closure derived by Dione et al; (2023) ranges -0.9 to -1.1. Can you explain this discrepancy?

Q1.5 Line 310-311: the alpha-closure derived for IOP14 (06:11 to 06:47) is 0.45, while the same alpha-closure derived by Dione et al; (2023) is 0.6. Can you explain this discrepancy?

Q1.6 What is the impact LWC0 measurement variability (from CDP and Visibility) on the results presented in Figure 5?

**Section 3.4** compares adiabaticity (definition 3) with equivalent adiabaticity (definition 2).

Alpha (definition 3) does not consider the fog layer at the top that is diluted by entrainment of dry air, hence it is expected to be larger than the equivalent alpha (definition 2) that accounts for the entire for LWP and depth.

Q1.7 What is the impact of the variability in the entrainment layer (depth, shape of LWC profile in the layer) on the comparison shown in Fig. 7 ?

**Major comment 2: thin-to-thick fog transition**

In Section 2.3, the thin-to-thick fog transition is presented as a time when the transition occurs. And thresholds are defined for five variables to identify the transition time linked to different processes that affect the transition: Net LW radiation, temperature gradient between 50 and 25m, TKE, CTH and LWP. A multi-parameter evaluation of the transition is interesting. However, the transition from thin to thick fog should rather be defined as a time interval with a beginning and an end (as proposed by Dione et al. 2023 using three parameters), rather than as an instant in time. Dione et al. (2023) shows that the duration of the transition phase is variable from one fog event to another.

Here, you propose the transition phase duration to be time interval between the first and the last thresholds of the five variables (L213). Later you mention "scattering" between thresholds (L216) and "period of uncertainty" (L324).

Q2.1 I suggest that you revise the definitions of thin-to-thick transition to include both thresholds for multiple variables, and time of onset of transition, and time of ending of the

transition when the fog has reached an adiabatic state. This would allow you to evaluate if multiple thresholds are reach in a short amount of time favours a rapid transition from thin to thick, and reversely if a slow transition can be caused by inconsistencies in the different processes involved in the transition.

**Q2.2** Could you propose a more thorough definition of thick-to-thin transition duration and explanation of the duration, based on the values reached by the five variables compared to the threshold that you identified.

L470 (Section 4.2.1) Explain how the time interval between first and last threshold can be compared with the transition phase duration of Dione et al. (2023) for IOP14

L512 (Section 4.2.2) You write "Surprisingly, reverse LWC trends were observed for profiles # 6 and # 7 with α <0 even though the thin-to-thick transition had already occurred one hour earlier."
In fact, this is not surprising for a slow transition. The fact that thresholds are reached does not mean that transition is over and that fog has become adiabatic.
Discuss how the LWC profiles temporal evolution that you show in can explain the slow transition.

L637, 638 "uncertainty periods" is used again for transition phase duration. Please revise.

**Minor comments.**

Section 2.1. Check consistency in the tense used throughout this section (past, present, future).

L106 Change "spread" to "distributed"

L119: CBH cannot be derived unambiguously with a cloud radar alone as the signal in the lower part of the cloud can come from cloud droplets or from drizzle below the cloud. CBH is usually derived from a ceilometer.

L129 "However, we analyse here all the data collected during the SOFOG3D experiment, and we then use independant retrieval." Not clear, please rephrase.
"independant" → "independent"

L137 "aspirates" → "sucks in"

Table 2. What objective criteria do you do define "radiative" vs "radiative-advective" fog types?

L203 "These discrepancies may be explained by the contrasting environment between the two measurement areas (Thornton et al., 2023)." Explain what contrasts you are referring to ?

L307 "These results hightlight that while the adiabatic model correctly represents thermodynamical and microphysical properties of well-mixed fogs, it does not represent the properties of optically thin fogs at all." Does anyone expect the adiabatic model to correctly represent optically thin fog ?

"hightlight" → "highlight"

L378 "significant values" → "large values"

L393 "This attests that differences observed from our dataset result from the actual properties of the fog sampled during SOFOG3D and not from the measurements used (in situ or remote sensing) to compute the fog adiabaticity from closure." Not clear, please rephrase or explain.

L634 Replace "if" by "while".

L703 "period of uncertainty" please rephrase according to Major comment 2 discussions.

L703 "more suited …" than what ?

L711 " larger negative values …" than what ?

---

## Author Response (AR1)

The authors would like to thank the reviewers for their helpful comments and the time they spent on reviewing this work. Below are the responses to all comments. Answers to the reviewers' comments appear in blue, while modifications in the article appear in red.

**Reviewer 1**

I found the paper generally well written and suitably laid out. There is a lot of information presented, and a significant number of figures, including those in the appendices. I did not find this a problem but there may be scope to shorten the paper a little, starting with the appendices. I consider the paper suitable for publication after minor corrections and alterations, but ask the authors to give my main comment serious consideration as I believe that the paper could be significantly improved from one that just reports on the morphology of fog, to one that provides new insight into fog dynamics.

Main comment

A major element of the paper is to report on the adiabaticity of the observed fog cases. Whilst some background on this parameter is given, I feel the presentation lacks focus and justification as to why studying adiabaticity is important. For example, what new insights into the physics of fog can we learn, and how can this be promulgated into improvements in NWP of fog?

It is true that we did not emphasize too much the interest of studying the adiabaticity of fog. Adiabaticity is the basic assumption in cloud physics to predict LWC in shallow non precipitating clouds from thermodynamics properties. It relies mainly on the parcel theory applied to a volume of air that cools as it rises in the atmosphere due to the pressure decrease. Adiabaticity, defined as the ratio of actual LWC (or LWP) to the adiabatic value, is a key parameter that describes the extent to which the actual liquid water amount in cloud deviates from the thermodynamically predicted adiabatic value. This parameter has been extensively studied in shallow non precipitating clouds thanks to aircraft measurements (Brenguier et al., 2011; Wood et al., 2012; Braun et al., 2018 and references therein, among many others). In such studies the adiabaticity ranges from 0 to the adiabatic value, and this parameter is mainly used to evaluate the reduction of LWC due to entrainment-mixing and precipitation processes.

Toledo et al (2021) used the adiabatic hypothesis to predict LWP evolution of fog layer from remote sensing instruments and visibility sensor at surface. In their conceptual model they add a term $LWC_0$ to account for the non-zero surface liquid water content (their Eq. (2)). They further introduced an equivalent adiabaticity $\alpha_{eq}$, defined as the constant adiabaticity value that would give the same LWP. They showed that when fog is developed (LWP > 30-40 g/m²), this adiabaticity parameter tends to ~0.66 indicating that fog is buoyant. However they noticed that thinner fog (LWP < 20 g/m²) has adiabaticity values below 0.6 and can even reach negative values.
A negative adiabaticity results mathematically from high $LWC_0$ values at the surface and low LWP of the fog layer. This reflects that the LWC profile is not increasing with height, and consequently that the basic assumption of the conceptual model is not valid.

To our knowledge adiabaticity has almost never been examined in fog layers due to the lack of in situ measurements. We then analyse our in situ vertical profiles of LWC to document the real trend of LWC evolution with altitude. We use the adiabaticity parameter to 1) compare with the results from Toledo et al (2021) that uses a remote sensing approach and 2) quantify the agreement of in situ data with the expected increase of the adiabatic assumption.

We show that the adiabaticity evolves from negative to positive values due to reversal of the LWC profile that initially decrease with altitude in stable conditions.

These findings will be used to evaluate the realism of numerical simulations of fog life cycle.

It's not clear however at this stage how this could be promulgated into improvements in NWP of fog. Indeed, the analysis of fog adiabaticity throughout its life cycle enables the discrimination between thin and thick fogs, but such a distinction is not implemented yet in microphysical schemes of numerical models.

We have made the following changes in the introduction :

L82: Adiabaticity is a key parameter that describes the extent to which the actual liquid water amount in cloud deviates from the thermodynamically predicted adiabatic value. It has been extensively studied in shallow non precipitating clouds thanks to aircraft measurements (Brenguier et al., 2011; Wood et al., 2012; Braun et al., 2018), but in situ observations of the vertical profile of fog microphysics are sorely lacking in the literature. Addressing this issue in fog layers is important to evaluate fog life cycle prediction models and to better understand the physical mechanisms underlying the transition from thin to optically thick fog to improve numerical weather simulations.

And in section 3.1 :

L337: Therefore, analysis of $\alpha_{eq}$ (CTH) enables to discriminate between optically thin and thick fog, characterized by negative and positive values, respectively.

L339 Note that negative adiabaticity values result mathematically from high $LWC_0$ values at the surface and low LWP of the fog layer. This reflects that the LWC profile is not increasing with height, and consequently that the basic assumption of the conceptual model is not valid.

I felt that the choice to reject data at cloud top and only evaluate adiabaticity over the lower to mid-fog region is a little arbitrary, and means that the results are not quantitative, and for indication only. I don't believe that the method used invalidates the conclusions made, but consider that a better analysis method could have provided more significant results. For example, the authors may have been able to study the main process creating sub-adiabatic liquid water profiles within adiabatic fog (rather than simply reporting on levels of adiabaticity). This is an important process that has great relevance to NWP of fog since it partly controls the amounts of liquid water in the fog. Clearly, entrainment at fog top is an important process in this regard. Whilst this is mentioned in the paper it is not studied and no attempt to quantify it is made. I suggest that, if possible, the authors consider extending their study to assess how the levels of relative humidity, wind shear and turbulence at fog top,

influence the levels of adiabaticity within adiabatic fog.

We fully agree with the reviewer that entrainment at fog top is a very important process that partly controls the available liquid water in the fog, and that the analysis of vertical profiles with reduced LWC layer at fog top would provide valuable insights to improve our understanding of fog life cycle that will benefit to NWP model.
In fact, that is exactly what we originally planned to do.

However, it appeared that during the SOFOG3D experiment, most of the LWC profiles were collected in thin fogs and they revealed that LWC rather decreases with altitude contrary to what is expected in a well-mixed buoyant fog layer. We then focused on the thin-to-thick transition and we found that LWC profiles evolve from reverse to increasing with height during the transition phase.
We then used the local adiabaticity as a parameter to contrast LWC vertical profiles between optically thin and thick fog and to quantify the change of the profile slope during the fog life cycle. We choose to remove the entrainment layer in order to avoid the decrease of the adiabaticity resulting from the sharp LWC decrease at the fog top. This methodology then allows to emphasize the contrast between thin and thick fog.

IOP 6 (Fig. 11e of the article) illustrates the relevance of calculating adiabaticity without taking the diluted layer into account. Indeed, vertical profile #8 exhibits LWC values close to the theoretical adiabaticity up to the base of the diluted layer at about 150 m height, resulting in an $\alpha$ value of 0.98, which is reduced to 0.8 if the diluted layer is taken into account. The next two profiles (between #8 and #9) also present $\alpha$ values close to 0.8, but $\alpha_{tot}$ is reduced to 0.57 and 0.15, respectively, when the diluted layer is included. In contrast, profile #9 shows a lower $\alpha$ value of 0.51, characterizing a dilution of the LWC throughout the fog layer and not restricted to the entrainment layer.
Therefore, evaluation of adiabaticity over the lower to mid-fog region appears to be a relevant metric by characterizing the overall deviation of the LWC profile from the adiabatic.
Obviously this methodology did not allow to examine the main processes creating sub-adiabatic conditions.

Comparison of $\alpha$ and $\alpha_{tot}$ is reported in Fig. 1 to evaluate the impact of removing the diluted layer at fog top on the adiabaticity parameter. This reveals that the reduction in adiabaticity when taking into account the diluted layer could be significant, but is not systematic.
Furthermore, it appears that the analysis of subadiabaticity throughout the fog life cycle is only possible for IOP 6c (blue dots), which with about 20 profiles has a sufficient number of observations. For the other cases the number of available profiles is too limited.
Note that out of the 140 profiles (127 in fog and 13 in stratus), only 42 (32 and 10) have adiabaticity parameter > 0.1, all others were performed in thin fog or diluted stratus. And their reverse profiles with maximum LWC values near the ground limit the possibility to examine the influence of entrainment at the fog top.

Extending the study to assess how the levels of relative humidity, wind shear and turbulence at fog top, influence the levels of adiabaticity within adiabatic fog is a very interesting suggestion. As mentioned above, this could only be achieved for IOP 6c. However, this is

not straightforward due to the large uncertainty of temperature and humidity measurements by in situ probes at the exit of the saturated fog layer and the limited sampling above the fog top, both of which prevent an accurate measurement of the thermodynamic profiles above the fog layer. Such an analysis would then require a substantial additional contribution that would lengthen the already overly long paper.
We therefore consider it more appropriate not to address this issue in this paper.

As a general comment, despite the significant efforts that have been made to sample the fog life cycle, more observations of the vertical structure of fog are therefore still needed, mainly for optically thick fogs, to conduct a comprehensive study of the fog top entrainment process.

[Figure]

Figure 1 : $\alpha$ as a function of $\alpha_{tot}$. Fog and stratus samples are indicated by dots and diamonds, respectively.

We have made the following changes in Sec. 3.2, 3.4 and Discussion :

L392: With such a procedure $\alpha = 0.63$ for IOP 14, close to $\alpha_{eq}$ (CTH), and the corresponding fit to the measurements better represents the general shape of the vertical profile than $\alpha^{closure}_{eq}$ and $\alpha^{closure}_{eqCDP}$. In addition, taking into account the entrainment-mixing layer above 220 m reduces strongly the adiabaticity to $\alpha_{tot} = 0.36$ (solid blue line).

L394: Thus, the local adiabaticity derived by truncating the entrainment-mixing layer for well-mixed buoyant fogs seems to be a relevant metric to characterize the global deviation of the LWC profile from the adiabatic one.

L501: Removing the diluted region at fog top for well-mixed fog increases $\alpha$ values. However only a few samples over the 32 profiles identified as increasing with height are significantly impacted and the comparison with $\alpha$ calculated over the entire profile leads to very similar results (not shown).

L700: As a result, it provides valuable information to investigate departure of the general trend of the profile from the adiabatic model, and to document the evolution during the fog life cycle, highlighting the contrast between optically thin and thick fogs.

**Other comment**

The term, 'Condensation rate', is used throughout the document but I am wondering if it is the most appropriate term since it implies dql/dt, whereas what is being talked about seems to be better described as the 'condensation amount'.
We use 'condensation rate' as it refers to dql/dz in Eq. (2) accordingly with the literature.

Line 111, 'SW to NE' : in which sense, clockwise or anticlockwise?
Corrected

Page 6. Table 1. The error quoted for the TKE appears to be an instrumental one. However, would you not expect the *random* error to be more like 20%?
We agree with the reviewer. TKE measurements were implemented using the eddy covariance method. Different approaches were conducted in the literature to estimate the random error uncertainty using an integral time scale (Lenschow et al, 1994) or including both autocorrelation and cross-correlation variance (Finkelstein and Sims, 2001), indicating a random error between 10 and 15 % (Fig. 2 of Finkelstein and Sims, 2001). In addition, Price. (2019) estimates an uncertainty of 20% on the measurement of the vertical velocity variance during the LANFEX fog campaign. Therefore, we modified in Tab. 1 the TKE uncertainty to 20% to account for the random error in addition to the instrumental error of about 1.5 %.

Figure 3 caption. Not all the lines and features of the figure are explained in the caption. Please also explain what is meant by 'local values'.
Corrected. We have modified 'local values' by 'local adiabaticity', which represents the vertical gradient of LWC and temperature measured by the CDP.

Lines 365-368. It appears to be assumed here that it is the Gultepe inversion that is inaccurate and that the measured $LWC_0$ is accurate. However, there is no discussion of the random errors in the measured $LWC_0$ to justify this assertion, or those from the visibility measurement used in the Gultepe inversion. I expect that $LWC_0$ is measured over a relatively short time scale (not specified, but perhaps a few seconds?). We know from many surface-based observations of fog water (using similar devices to the one used here), that there is rapid variation in Liquid water content on timescales of seconds to minutes. Therefore, a short sampling period will be prone to larger random errors, meaning that the assumption here that the measurement is 'accurate' may be erroneous and in fact the Gultepe inversion may not be as inaccurate as assumed here. I think that this conclusion should be re-evaluated once random errors for the measured $LWC_0$ and visibility has been provided. Also, in this regard, when short time scales are used, it is important that the two instruments are closely co-located to meaningfully compare the results between them.

We fully agree that random errors introduce significant variability that must be taken into account when comparing measurements, and that such information was lacking.

CDP measurements of $LWC_0$ are determined as the median value at the lowest altitude range, representing from 7 to 17 1s samples, thus indeed measurements on a very short time scale, typically a tens of seconds. We computed distribution for each vertical profile and use 1st and 3rd quartiles to assess the variability during the corresponding time interval.
The corresponding intervals have been added on the Fig. 5 of the paper (and similarly on the Fig. 4 as well).
Comparison of $LWC_0$ values from the CDP and visibility measurements with their variability intervals, reported in Fig. 5, reveals that the variability of the CDP measurements is obviously larger than that of the visibilimeter, but it is not large enough to explain the observed discrepancies. However, this does not exclude the possibility that the differences come from temporal fluctuations at larger scales because this would have required a longer series of measurements.

As pointed out by the reviewer, co-location of measurement sites are also essential. Unfortunately, due to failure of the visibilimeter deployed at the Charbonnière site, co-located measurements were only possible for 5 IOPs out of the 12 studied, representing 37 samples out of 140, i.e. about 26% of the profiles. As for the 7 other IOPs, the visibilimeter at the Jachère site (1.4 km away) must be used. Comparison of LWC with a Fog Monitor deployed at the same site shows a better overall agreement but with similar dispersion.
It follows that inaccuracies of the Gultepe et al. (2006) parameterization explain some of the differences observed with the CDP, but not as much as we previously concluded. The text has been therefore modified accordingly.

Figures 4 and 5 of the paper have been modified : variability intervals have been added to represent the uncertainties due to time scale and differences in location.

We have made the following changes in Sec. 3.3, discussion, conclusion and abstract :

L402: For each vertical profile, $\alpha_{eq}$ is computed from Eq. 6 by using the median values over the duration of the profile of CTH from the radar, LWP from the MWR and visibility measurements. Due to an instrument failure, data from the visibilimeter deployed at the Charbonnière site were only available for IOPs 9a to 13b. As for the other 7 IOPs, the visibilimeter at the Jachère closure site (at 1.4 km) must be used. To compute the fog adiabaticity from closure with CDP data, $\alpha^{closure}_{eqCDP}$, CTH is determined by using an LWC threshold of 0.01 g.m$^{-3}$, the LWP is calculated by integrating the median value of LWC in each altitude range up to CTH, and $LWC_0$ is defined as the median value of LWC in the lowest altitude range. The median height corresponding to $LWC_0$ is 3.5 m with 1st and 3rd quartiles of 2.5 m and 6.2 m, respectively, i.e. very close to the height of the visibilimeters deployed at 3 m above ground level, but over a short time period typically of about ten seconds. Except for $CTH_{CDP}$, 1st and 3rd quartiles of the distribution of each parameter are calculated for each vertical profile, to assess the variability during the corresponding time interval.

L440  Part of the discrepancies could arise from temporal fluctuations of $LWC_0$ during the time taken by the balloon to complete each profile, which cannot be truly inferred from the CDP measurements. It is also likely that the use of the visibility measurement from the Jachère site for 7 out of the 12 IOPs studied contributes to the dispersion. Therefore, these sensitivity tests reveal that the discrepancies between $\alpha^{closure}_{eq}$ and $\alpha^{closure}_{eq\ CDP}$ arise both from the $LWC_0$ values derived from visibility measurement and by differences in LWP between the CDP and the MWR.

L748: We have shown that large differences exist between $LWC_0$ values derived from the CDP and from the visibility by the parameterization of Gultepe et al. (2006), that contribute to the uncertainty in the calculation $\alpha^{closure}_{eq}$ .

L809 and 12: The comparison shows a large variability resulting from differences between both LWC values at ground and measurements of LWP., [...].

*Line 387, you mean 'negative'?*
The sentence has been modified as follows :

L460: This limit has been extended here to 10 m (dashed line) to compare with in situ data : most of them are situated above this curve reflecting that positive values of $\alpha^{closure}_{eq\ CDP}$ can be observed for CTH lower than 85 m.

*Line 408. The averaging time for the CDP for each reported sample can be set manually. What was it set to?*

The 1 Hz CDP measurements were not averaged. We calculate statistics on 1 Hz samples in each altitude range. A threshold of 5 samples per altitude range, except for optically very thin fogs (< 10 m), was nevertheless applied to ensure minimum representativeness.

The sentence has been modified as follows :
L491: Except for very thin fog episodes, the minimum number of 1 Hz CDP samples required to calculate the statistic over each altitude range was set to 5.

*Line 446. Units for lapse rate?*
$\gamma$ and $\alpha$ are unitless, defined as the ratio between the local adiabaticity and the theoretical adiabaticity ($\Gamma$ad and $\Gamma$w respectively). The end of Sec. 3.2 has been rephrased to clarify this point.

L387: To better estimate the agreement of the measurements with the adiabatic model, we calculate the unitless local adiabaticity $\alpha$ and $\gamma$ with respect to LWC and lapse rate, respectively, by determining the slope of the linear regression between medians values of the statistics at each altitude range and dividing by $\Gamma_{ad}$ and $\Gamma_w$.

*Figure 8 caption. Upward and downward radiation explanation is incorrect.*
Corrected

*Line 491. Suggest not using 'supersite' and sticking to Jachere or Charbonniere.*
Corrected.

*Line 522. Any suggestions as to why the actual evolution did not match that expected?*
The assumption is that we observe reversed vertical profiles of LWC (maximum LWC values at ground level decreasing with height) after the transition to optically thick fog during profiles # 6 and # 7 due to sedimentation and collision-coalescence processes that redistribute the LWC through the fog layer from the top to the ground, discussed in Sec. 5.

The sentence has been modified as follows :

L612: Consequently, it is clear from Fig. 10f that the temporal evolution of $\alpha$ and $\gamma$ during the fog life cycle follows the same trend as for IOP 14, from negative values when the fog is optically thin to positive values after the transition to thick fog, except for the sharp decrease in $\alpha$ for profiles # 6 and # 7, which disrupts the overall trend, due to sedimentation and collision-coalescence processes as discussed in Sec 5.

*Line 528. I can't see the upper liquid layer mentioned in fig. 11c. Could you make this clear?*
We wanted to emphasize that the formation of the fog was almost instantaneous with first radar echoes located at altitude but effectively it is not clear.
L620: We replaced 'upper liquid water layer' by 'a 75 m thick liquid water layer'.

*Line 635. Why should geometric depth be more important than LWP as a driver for the transition to adiabatic fog ?*

We suggested that the geometrical depth is the main driver because we interpreted our observations as showing the existence of a single threshold in CTH while there would be different thresholds in LWP depending on fog characteristics (SIRTA or SOFOG3D). And also because the adiabaticity is parameterized as function of CTH rather than LWP.
But this suggestion is inconsistent with LWP being the main contribution to optical depth in the LW radiations, compared to CTH (Wærsted et al., 2017). And this threshold of 110 m for CTH is indeed consistent with our parameterization of the adiabaticity with $\alpha^{closure}_{eq\ CDP}$ (CTH) > 0.5 when CTH > 110 m corresponding to the end of the transition.

We therefore replaced the sentence with the following, and deleted this assertion in the conclusion.

L730: This is confirmed by the new parameterization of $\alpha^{closure}_{eq\ CDP}$ (CTH) which exceeds 0.5 when CTH reaches 110 m.

*Line 650. See point 5.*
Corrected, we rephrased as follows:

L748: We have shown that large differences exist between $LWC_0$ values derived from the CDP and from the visibility parameterization of Gultepe et al. (2006), that contribute to the

uncertainty in the calculation of $\alpha^{closure}_{eq}$

*Line 670. Did you examine the wind and temperature profile in detail for this case? It may be possible that differential advection had mixed two different fog regimes, one over the other, which might explain your observations.*

We analyzed the wind and temperature profiles and agree that these observations support this hypothesis of differential advection that could explain the two-layer structure of profile #5. We have therefore added this sentence :

L769: The wind and temperature profiles reported in Dione et al. (2023) (their Fig. 8d) indicate wind speed shear and strong cooling after 2300 UTC below 200 m height, suggesting that differential advection could explain this two-layer structure.

**Reviewer 2**

This paper studies vertical profiles of liquid water content (LWC) observed in fog layers during a field campaign carried out in southwestern France. The twelve fog events sampled during the campaign concern "thin" fogs occurring in stable conditions, characterized by decreasing LWC and increasing temperature as a function of altitude and vertically developed (or "thick") adiabatic fog, characterized by increasing LWC and decreasing temperature as a function of altitude.

The work is original in that it relies on a comprehensive set of in-situ measurements of fog microphysics, complemented by remote sensing using a cloud radar and microwave radiometer.

The paper includes several interesting investigations that make it worthwhile to consider this paper for publication in Atmospheric Physics and Chemistry: Vertical profiles of LWC (measured by a cloud droplet probe, CDP) and temperature are analysed at different stages of the fog life cycle. The conditions that lead to thin-to-thick transition are investigated for four fog events using remote sensing measurements, surface measurements and in-situ vertical profiles. This is important to evaluate how the representation of transition can be improved in model simulations. Further, this analysis is compared to a conceptual model of fog adiabaticity proposed and exploited earlier in the literature (Toledo et al. 2021 and Dione et al. 2023) and provides a detailed assessment of the performance of this conceptual model.

The paper is clearly written and well organized. However, two topics require thorough attention and revisions, while others are more minor.

*Major comments*

**Major comment 1: derivation and uncertainty of fog adiabaticity.**

Fog adiabaticity is studied extensively in the paper, both from in-situ LWC measurements

and using the conceptual model proposed by Toledo et (2021). The paper presents several methods to derive fog adiabaticity.

1. Equivalent adiabaticity from Closure (Eq. 5), where input variables are derived from remote sensing measurements and horizontal visibility
2. Equivalent adiabaticity from Closure (Eq. 5), where input variables are derived from in-situ CDP measurements
3. Adiabaticity from fitting LWC profiles from the ground to just below the diluted layer.

**Section 3.2 and 3.3** are dedicated to comparing the first two definitions,

The balloon ascent (that carries the CDP sonde) takes 15-40 minutes to cover the vertical extent of the fog layer, depending on its depth. In the equivalent adiabaticity formulation (Eq. 5), there is an inherent hypothesis that the LWC0 measured at the surface is consistent with the integral of LWC (the liquid water path, LWP) and the fog layer depth (provided by CTH). We know that CTH, LWP and LWC_0 can vary significantly in 20-30 minutes.

Q1.1 Hence what is the uncertainty in deriving equivalent adiabaticity (definition 2) from CDP measurements given the temporal variations of LWC during the ascent or descent of the balloon?

We agree with the reviewer that uncertainties on the CDP measurements, in particular in $LWC_0$, are important to consider when calculating equivalent adiabaticity from Closure. This lack in the manuscript was also pointed out by reviewer 1.

To assess the variability of each parameter during a vertical profile, we computed the distributions during the corresponding time interval and used 1st and 3rd quartiles to represent the uncertainty. The corresponding intervals have been added on the Fig. 4 and 5 of the paper.
This method works well for remote sensing instrument and horizontal visibility (definition 1) but obviously not for CDP measurements below the tethered balloon because of the displacement of the instrument.

As explained to Reviewer 1, CDP measurements of $LWC_0$ are determined as the median value at the lowest altitude range, representing typically a tens of seconds of measurements. We have checked however that the dispersion between $LWC_0$ values from the CDP and visibilimeter measurements reported in Fig. 5, is not much larger than those obtained by a ground based FM120 at the Jachère site, indicating that $LWC_0$ measurements from the CDP determined over a short period of time are consistent with those from the visibilimeter over the duration of the profile.

For the LWP derived from CDP measurements, the variability is estimated by integrating the 1st and 3rd quartiles of the LWC distribution calculated at each altitude level. It is interesting to note that comparison of LWP derived from the CDP and measured by the MWR from Martinet et al. (2025), in prep, shows that this spatial variability with altitude is very similar to the temporal variability of the LWP from MWR measurements during the profile time period. And that the resulting uncertainty on the derived value of adiabaticity by closure is small compared to the overall dispersion of the samples (Fig. 4b).

We have made the following changes in Sec. 3.3 :

L405: To compute the fog adiabaticity from closure with CDP data, $\alpha^{closure}_{eq\ CDP}$, CTH is determined by using an LWC threshold of 0.01 g.m$^{-3}$, the LWP is calculated by integrating the median value of LWC in each altitude range up to CTH, and $LWC_0$ is defined as the median value of LWC in the lowest altitude range.

L409: Except for $CTH_{CDP}$, 1st and 3rd quartiles of the distribution of each parameter are calculated for each vertical profile, to assess the variability during the corresponding time interval.

L433: Note that the variability represented by 1st and 3rd quartiles is relatively small compared to the dispersion and similar for both estimates, indicating that the temporal variability of LWP during the profile does not have a significant impact on the derived value of adiabaticity by closure.

Q1.2 What temporal smoothing or averaging is used for CTH, LWP and visibility to derive equivalent adiabaticity (definition 1).

CTH, LWP and visibility observations used to derive the equivalent adiabaticity are determined as the median values over the duration of the profile inside fog. We have modified the beginning of section 3.3 :

L402: For each vertical profile, $\alpha^{closure}_{eq}$ is computed from Eq. 6 by using the median values over the duration of the profile of CTH from the radar, LWP from the MWR and visibility measurements.

L409: Except for $CTH_{CDP}$, 1st and 3rd quartiles of the distribution of each parameter are calculated for each vertical profile, to assess the variability during the corresponding time interval.

Q1.3 What is the impact of these uncertainties on the comparisons of the two alpha-closures made in figure 4?

The uncertainties on the equivalent adiabaticity have been estimated by the variabilities represented by 1st and 3rd quartiles of the distribution of each parameter for each profile, and are superimposed on Fig. 4 and Fig. 5.
As a general comment, it appears that the resulting uncertainties are rather small compared to the overall dispersion of the samples.

Obviously we underestimated the actual uncertainties resulting from the CDP measurements because, as already mentioned above (Q1.1 section), it is not possible to derive it, due to balloon displacement.
But this first study reveals that despite a large dispersion, the adiabaticity from closure values derived from in situ measurement through the fog layer are consistent with the ones derived from remote sensing and horizontal visibility observations. And that negative values

correspond in fact to reverse profiles with LWC decreasing with height. For such profiles with low LWP values we found that uncertainties on LWP measurements and $LWC_0$ retrievals induce inconsistencies closure between LWP, CTH and $LWC_0$ values, which implies that the closure conditions required for the calculation of $\alpha^{closure}_{eq}$ are not satisfied.
This is also the case when the sedimentation process redistribute the water inside the fog layer, increasing strongly $LWC_0$ values at the surface.

Fig. 4 and 5 have been modified.

L383: These observations indicate that for optically thin fogs characterized by low LWP values (LWP < 10 $g.m^{-2}$ ), uncertainties on LWP measurements and $LWC_0$ retrievals induce inconsistencies between LWP, CTH and $LWC_0$ values, which implies that the closure conditions required for the calculation of $\alpha^{closure}_{eq}$ are not satisfied.

L426: For this parameter, for which no variability from the CDP measurements can be calculated, the worst cases, from IOP 9b and 13b are mainly due to overestimation of CTH by the radar for actual CTH just above its detection limit that results in overestimation of $\alpha^{closure}_{eq}$.

L433: Note that the variability represented by 1st and 3rd quartiles is relatively small compared to the dispersion and similar for both estimates, indicating that the temporal variability of LWP during the profile does not have a significant impact on the derived value of adiabaticity by closure.

Q1.4 Line 310-311: the alpha-closure derived for IOP11 (22:09 to 22:23) is 0.53, while the same alpha-closure derived by Dione et al; (2023) ranges -0.9 to -1.1. Can you explain this discrepancy?

We calculated ($\alpha^{closure}_{eq}$) value by using LWP, CTH and $LWC_0$ measured values by applying Eq. 6 of Toledo et al. (2021), whereas Dione et al. (2023) determined $\alpha_{eq}$ values from the parameterization as function of CTH ($\alpha_{eq}$ (CTH))  (Eq. 7 of Toledo et al. (2021)). Such a discrepancy illustrates the differences that can exist between a specific value under given conditions and a parameterization that represents the overall trend of the parameter.

We have added the following sentence to emphasize the differences in terms of methodology between these two studies to compute $\alpha_{eq}$.

L367: It is worth noting that these values differ from the study by Dione et al. (2023) who computed $\alpha^{closure}_{eq}$ based on a parameterization in CTH.

Moreover, $\alpha_{eq}$ (CTH) values for IOP 11 and 14 have been added on Fig. 3c and 3a of the article in dotted blue lines, respectively, and theoretical information on $\alpha_{eq}$ (CTH) following Toledo et al. (2021) have been added at the end of Sec. 3.1 :

L332 : They thus proposed the following parameterization :

$\alpha_{eq}$ (CTH) = $\alpha_0$ (1 − $e^{-CTH-HoL}$) (7)

where $\alpha_0 := 0.65$, $H_0 = 104.3$ m and $L = 48.3$ m.

Dione et al. (2023) used this parameterization to analyze the four deepest fogs of SOFOG3D. They revealed negative values of $\alpha_{eq}$ (CTH) during the stable phase, which increase from 0 to 0.5 during the thin-to-thick transition and finally exceeds 0.5 when LWP > 20 g.m$^{-2}$ .Therefore, analysis of $\alpha_{eq}$ (CTH) enables to discriminate between optically thin and thick fog, characterized by negative and positive values respectively.

Note that negative adiabaticity values result mathematically from high $LWC_0$ values at the surface and low LWP of the fog layer. This reflects that the LWC profile is not increasing with height, and consequently that the basic assumption of the conceptual model is not valid.

Regarding IOP 11 when the fog is optically thin, vertical profile #2 between 2209 and 2223 UTC was sampled when the fog layer was thinning, characterized by CTH values decreasing to about 40 m (Fig. 10c of the article), low LWP values (LWP < 10 g.m$^{-2}$) and $LWC_0$ above 0.1 g.m$^{-3}$. This results in a positive $\alpha^{closure}_{eq}$ when applying Eq. 6 of Toledo et al. (2021) with a value of 1.03, and a negative $\alpha_{eq}$ (CTH) value of -1.94 when applying Eq. 7 of Toledo et al. (2021), in agreement with the negative values observed by Dione et al. (2023) and with the general trend of LWC with height. Therefore, these observations suggest that for optically thin fogs characterized by low LWP values (LWP < 10 g.m$^{-2}$), uncertainties on LWP measurements and fluctuations of $LWC_0$ induce inconsistencies between LWP, CTH and $LWC_0$ values, which implies that the closure conditions required for the calculation of $\alpha^{closure}_{eq}$ are not satisfied.

L382: In addition, $\alpha_{eq}$ (CTH) reveals a negative value of -1.94 in agreement with $\alpha^{closure}_{eqCDP}$ and the general trend of LWC with height, related to a low CTH value of 55 m. These observations indicate that for optically thin fogs characterized by low LWP values (LWP < 10 g.m$^{-2}$), uncertainties on LWP measurements and fluctuations of $LWC_0$ retrievals induce inconsistencies between LWP, CTH and $LWC_0$ values, which implies that the closure conditions required for the calculation of $\alpha^{closure}_{eq}$ are not satisfied.

Q1.5 Line 310-311: the alpha-closure derived for IOP14 (06:11 to 06:47) is 0.45, while the same alpha-closure derived by Dione et al; (2023) is 0.6. Can you explain this discrepancy?

As for IOP 14 after thin-to-thick transition, CTH derived from the BASTA radar during profile #6 between 0611 and 0647 UTC results, using the CTH parameterization of Toledo et al. (2021), in an $\alpha_{eq}$ (CTH) value close to the 0.65 asymptote, while considering LWP and $LWC_0$ values reduces $\alpha^{closure}_{eq}$ close to 0.48.

L375: The profile of $\alpha_{eq}$(CTH), represented by a dotted blue line, enables to retrieve an equivalent adiabaticity more in agreement with the general trend of LWC with altitude. The value of 0.64 is close to the asymptote of the parameterization ( $\alpha_0 = 0.65$) due to a high CTH reaching 255 m.

In addition, we propose in Sec. 3.3 a new parameterization of $\alpha_{eq}$ (CTH) on the basis of the SOFOG3D observations using the same methodology as Toledo et al. (2021), since on Fig. 6b most of samples are located above the previous parameterization curve based on observations at the SIRTA site. However, due to the lack of observations of deep optically

thick fogs, the asymptotic value of 0.65 retrieved by Toledo et al. (2021) has been kept. Figure 6c has been added to illustrate the methodology applied to derive the new parameterization, which extends the previous parameterization to optically thin fogs.

L462: Statistics on classes of 20 m width indicate that the median of $\alpha^{closure}_{eq\ CDP}$ values is positive when CTH > 70 m (Fig. 6c).

L469: Following Toledo et al. (2021) a parameterization of $\alpha_{eq}$ as a function of CTH is derived by minimizing Eq. 7 with respect to the median value of $\alpha_{eq}$ for each CTH range of 20 m. Only intervals with more than 5 valid samples are used. Given the lack of valid samples for CTH > 140 m (Fig. 6c, green boxplots), the $\alpha_0$ asymptotic value is set to 0.65 as determined by Toledo et al. (2021). Thus, only the parameters $H_0$ and $L$ from Eq. 7 are determined, which represent the typical CTH values at which LWC increases with height ($\alpha_{eq}$ = 0) and at which the thin-to-thick transition begins ($\alpha_{eq}$ = 0.5), respectively. The retrieved values, $H_0$ = 65.3 m and $L$= 34.8 m, are lower than the values obtained by Toledo et al. (2021), 104.3 and 48.3 m, respectively. Therefore, the thinner fog layers sampled during SOFOG3D suggest that fog become optically thick at lower CTH than previously derived from SIRTA observations.

L480: Additionally, in situ data allow to characterize large negative $\alpha^{closure}_{eq}$ values relative to thin fogs, for which a new parameterization as a function of CTH has been proposed

A comparison between $\alpha$ and the new parameterization $\alpha_{eq}$ (CTH)$_{CDP}$ for the 127 vertical profiles has also been added in Fig. 7b. It shows that statistically, $\alpha^{closure}_{eq}$ appears to be a better predictor of fog adiabaticity than $\alpha_{eq}$ (CTH) since the parameterization does not allow to retrieve positive adiabaticity values when it is observed for optically thin fogs, and also because minimum values of $\alpha_{eq}$ (CTH)$_{CDP}$ remained larger than -3 that are clearly overestimated compared to $\alpha$.and $\alpha^{closure}_{eq\ CDP}$.

L499: Comparison of $\alpha$ with $\alpha^{closure}_{eq\ CDP}$ (Eq. 6) and $\alpha_{eq}$ (CTH)$_{CDP}$ from the new parameterization, are reported on Fig. 7 for the 127 selected profiles.

L508: In contrast, given that CTH only reaches 30 m, the parameterization leads to a negative value of $\alpha_{eq}$ (CTH)$_{CDP}$ (Fig. 7b). Indeed, as a general trend, the parameterization generates too much negative adiabaticity values due to low values of CTH, while $\alpha^{closure}_{eq\ CDP}$ are positive in agreement with $\alpha$. In contrast for thinnest layers, $\alpha_{eq}$ (CTH)$_{CDP}$ remained larger than -3 that are clearly overestimated compared to $\alpha$ and $\alpha^{closure}_{eq\ CDP}$ .

L515 Therefore $\alpha^{closure}_{eq}$ appears to be a better predictor of fog adiabaticity, with an $R^2$ value of 0.82, since the parameterization taking only CTH into account does not allow to retrieve positive adiabaticity values of optically thin fogs, and overestimates minimum values of thinnest fogs, leading to an $R^2$ value of 0.41.

Finally, a sentence summarizing the $\alpha_{eq}$ (CTH) study has been added to the abstract and conclusion and background is given in the introduction.

Q1.6 What is the impact LWC0 measurement variability (from CDP and Visibility) on the results presented in Figure 5?

As mentioned previously (Q1.1 page 11 and answers to Reviewer 1 comment page 6), 1st and 3rd quartiles have been added in Fig. 5 to represent the measurement variability from CDP and visibility. The resulting uncertainty in $LWC_0$ is larger from the CDP measurements than that of visibility. But except for a few samples, it is rather low and can not explain the overall dispersion of the samples. However, it is still possible that part of the discrepancies arise from temporal fluctuations of $LWC_0$ during the time taken by the balloon to complete each profile, which cannot be truly inferred from the CDP measurements. It is also likely that the use of the visibility measurement from the Jachère site for 7 out of the 12 IOPs studied, due to a failure of the visibilimeter deployed at the Charbonnière site, contributes to the dispersion.

It follows that inaccuracies of the Gultepe et al. (2006) parameterization explain some of the differences observed with the CDP, but not as much as we previously concluded. The text has been modified accordingly.

L440: Part of the discrepancies could arise from temporal fluctuations of $LWC_0$ during the time taken by the balloon to complete each profile, which cannot be truly inferred from the CDP measurements. It is also likely that the use of the visibility measurement from the Jachère site for 7 out of the 12 IOPs studied contributes to the dispersion. Therefore, these sensitivity tests reveal that the discrepancies between $\alpha_{eq}$ and $\alpha^{closure}_{eq}$ arise both from the $LWC_0$ values derived from visibility measurement and by differences in LWP between the CDP and the MWR.

**Section 3.4** compares adiabaticity (definition 3) with equivalent adiabaticity (definition 2).

Alpha (definition 3) does not consider the fog layer at the top that is diluted by entrainment of dry air, hence it is expected to be larger than the equivalent alpha (definition 2) that accounts for the entire fog LWP and depth.

_Q1.7 What is the impact of the variability in the entrainment layer (depth, shape of LWC profile in the layer) on the comparison shown in Fig. 7 ?_

As mentioned in the response to reviewer 1 (page 3-4), there are only 42 vertical profiles (32 in fog and 10 in stratus) that were identified as increasing with height and for which $\alpha$ was therefore recalculated on the lower part of the profile. Figure 2 shows that difference between $\alpha$ and $\alpha_{tot}$ increases with the relative thickness of the entrainment zone that reaches up to 50%.

Given the limited number of samples and the many sources of uncertainty on the different parameters, we did not examine in more detail the properties of the dilution zone that would allow to precisely assess the impact on the comparison with $\alpha^{closure}_{eq\ CDP}$ .

Indeed, for the 127 vertical profiles in fog, considering the values of $\alpha$ or $\alpha_{tot}$ in the comparison with the values of $\alpha^{closure}_{eq\ CDP}$ reveals little impact (Fig. 3a), the slope being slightly lower (1.45 compared with 1.52 in Fig. 7 of the article). Furthermore, focusing on the 32 profiles with LWC increasing with height (defined as $\alpha_{tot} > 0.1$) in fog, the comparison shows that most $\alpha_{tot}$ values are rather similar to those of $\alpha^{closure}_{eq\ CDP}$ , except a few profiles sampled during IOP 2a and 6c (Fig. 3b).

We have made the following changes in Sec. 3.4 :

L501: Removing the diluted region at fog top for well-mixed fog increases α values. However only a few sampled over the 32 profiles identified as increasing with height are significantly impacted and the comparison with α calculated over the entire profile leads to very similar results (not shown).

[Figure]

Figure 2 : α as a function of $\alpha_{tot}$. Colour correspond to the relative thickness of the entrainment layer. Fog and stratus samples are indicated by dots and diamonds, respectively.

(a)                                                                  (b)

[Figure]

Figure 3 : $\alpha_{tot}$ as a function of $\alpha^{closure}_{eq\,CDP}$ (a) for all vertical profiles in fog (b) for the 32 profiles with increasing LWC ($\alpha_{tot} > 0.1$).

**Major comment 2: thin-to-thick fog transition**

In Section 2.3, the thin-to-thick fog transition is presented as a time when the transition occurs. And thresholds are defined for five variables to identify the transition time linked to different processes that affect the transition: Net LW radiation, temperature gradient between 50 and 25m, TKE, CTH and LWP. A multi-parameter evaluation of the transition is interesting. However, the transition from thin to thick fog should rather be defined as a time interval with a beginning and an end (as proposed by Dione et al. 2023 using three parameters), rather than as an instant in time. Dione et al. (2023) shows that the duration of the transition phase is variable from one fog event to another.

Here, you propose the transition phase duration to be the time interval between the first and the last thresholds of the five variables (L213). Later you mention "scattering" between thresholds (L216) and "period of uncertainty" (L324).

**Q2.1** I suggest that you revise the definitions of thin-to-thick transition to include both thresholds for multiple variables, and time of onset of transition, and time of ending of the transition when the fog has reached an adiabatic state. This would allow you to evaluate if multiple thresholds are reach in a short amount of time favours a rapid transition from thin to thick, and reversely if a slow transition can be caused by inconsistencies in the different processes involved in the transition.

This is absolutely true that the notion of transition refers more to a temporal duration than to a given instant. In the article, the 5 thresholds defined on CTH, LWP, $LW_N$, $Grad_T$ and TKE respectively, correspond to the end of the transition time from optically thin to thick fog. No threshold was applied to determine the onset of the thin-to-thick transition due to the complexity of the case studies and the lack of thresholds defined in the literature. Following the approach of Dione et al. (2023) who proposed thresholds based on the values of $\alpha^{closure}_{eq}$, dT/dz and dCTH/dt to define the onset and end of the thin-thick transition, we propose in this article to refer to the analysis of Dione et al. (2023) for the transition onset, and to keep the 5 thresholds defined in the article to determine the end of the thin-thick transition. The time interval between the 5 thresholds indeed provides information on the uncertainty associated with the estimation of the end of the transition.

We have made the following changes in Sec. 2.3, and modified Table 2 accordingly.

L198: We choose to define the onset of the transition phase on the basis of the Dione et al. (2023) study. The transition phase ending is determined by applying the following five conditions for the four thick fogs, in order to assess the uncertainty associated with the definition of the transition phase duration. This methodology allows us to evaluate whether multiple thresholds reached in a short period of time result in a rapid transition phase duration, and whether large discrepancies on the ending of the transition phase can be caused by local or non-local processes.

Observations of the thin-to-thick transition period during the 4 deepest IOPs reveal no obvious tendency between the time interval at which the different thresholds are reached and the duration of the transition period. Indeed, IOP 11 has the lower uncertainty between all the thresholds (25 minutes) but a transition duration of about 1h30-2h. In the other hand

the uncertainty is higher for IOP 14 (48 minutes) but the transition period is much shorter < 1h. IOP 6 reveals strong discrepancies between all the thresholds (4h05) although the thin-to-thick transition occurs shortly after fog formation. And finally for IOP 9 the uncertainty period is also quite long (2h29) despite the fog is immediately thick. Therefore, multiple thresholds reached in a short period of time do not appear to favour a rapid thin-to-thick transition, and conversely, a slow transition is not systematically associated with inconsistencies between all thresholds.

We modified the text accordingly throughout the Sec. 2.3 and added the following sentence as conclusion :

L272: As a result, no tendency is observed between the time interval during which the different thresholds are reached and the duration of the transition period. This suggests that multiple thresholds reached in a short period of time do not seem to favour a rapid thin-to-thick transition and, conversely, that a slow transition is not systematically associated with inconsistencies between all the thresholds.

**Q2.2** Could you propose a more thorough definition of thick-to-thin transition duration and explanation of the duration, based on the values reached by the five variables compared to the threshold that you identified.

According to the answer to Q2.1, we propose to define the thin-to-thick transition duration based on the study by Dione et al. (2023) for the transition onset, and using the 5 thresholds for the transition ending. This methodology allows us to estimate an uncertainty on the transition period based on the differences observed on the transition ending times. Tab. 2 has been modified to include both the transition phase duration and transition ending time uncertainty.

In addition, the CDP measurements can also be used to define the optical depth τ for each of the 140 vertical profiles, defined as the integral of the extinction over the entire fog layer, which allows to discriminate between optically thin and thick fogs. We then added these values on the time series of Fig. 2, A1 and A2 of the article, corresponding to IOP 14, 11 and 6, respectively.

Analysis of optical depth temporal evolution during IOP 14 (Fig. 2g) reveals a significant contrast between the first 3 vertical profiles (τ < 2) when the fog is optically thin and the next 7 profiles after 0050 UTC (up to values of 8 after the dissipation in stratus) after the ending of the thin-to-thick transition (determined between [2342-0030] UTC (Tab. 2) and 0100 UTC in Dione et al. (2023)). For IOP6 (Fig. A2g), the optical depth follows a similar evolution with values close to 2 shortly after fog formation, decreasing to 0.32 and 0.21 for profiles # 4 and # 5, respectively, around 2300 UTC when the vertical development is stopped, and increasing again after midnight during the second vertical development, with values up to 6. A similar sharp increase is not observed for IOP 11 for which τ remains below 2 for the first 6 vertical profiles and only increases significantly after the dissipation in stratus with values close to 8 (Fig. A1g). But this can be explained by the occurrence of sedimentation and collision-coalescence processes that redistribute the LWC through the fog layer from the top to the ground, just after the end of the transition. Therefore, analysis of the optical thickness

allows to relate this parameter to the complex microphysical properties observed at the thin-to-thick transition of IOPs 6,11 and 14.

Optical depth parameter was then added in Sec. 2.3 with a threshold of 5 as proposed by Wærsted et al. (2017) to discriminate optically thick fogs. We find this threshold too high for our dataset, and we propose instead a value of 2 as more appropriate to discriminate between optically thin and thick fogs. We find that this complementary analysis provides convincing evidence on the characterization of the transition period, but of course optical depth derived from CDP measurements can not be used to provide an accurate estimation of the transition ending time because measurements are restricted to vertical profiles performed by the balloon and are thus not continuous in time.

We have made the following changes in section 2.3, and subsequently throughout the article, to include the transition phase duration instead of the transition time, adding the threshold on the optical depth.

L229: The optical depth, $\tau$ , increases as the fog becomes optically thick (Vehil et al., 1989). Wærsted et al. (2017) consider opaque fog when $\tau > 5$. Droplet size distribution measurements from the CDP collected during vertical profiles provide the opportunity to compute the optical depth of the fog layer as :

$$\tau = SUM \_Zb \wedge Zt \; \sigma_{ext} \, (z)dz.$$

where $\sigma_{ext}$ is the extinction, and Zb and Zt are the cloud base and cloud top heights, respectively. Optical depth from CDP measurements is computed for each of the 140 vertical profiles to provide an independent assessment of the optically thin and thick foggy periods.

L238: Optical depth (Fig. 2g) reveals low values for the first 3 profiles with $\tau < 1$, while it remains close to or above the threshold of 5 from profile # 4. This suggests that fog becomes optically thick between 2315 (# 3) and 0049 UTC (# 4). These observations are consistent with Dione et al. (2023) who defined the onset of the transition phase at 2330 UTC.

L257: The optical depth follows a similar evolution with values close to 2 shortly after fog formation, decreasing to 0.32 and 0.21 for profiles # 4 and # 5, respectively, around 2300 UTC when the vertical development is stopped, and increasing again after midnight during the second vertical development, with values up to 6. The threshold of 5 seems then too high since values are closer to 2 when fog is optically thick shortly after fog formation. Dione et al. (2023) defined the onset of the transition phase at 0000 UTC, which actually corresponds to the second period of thick fog. Consequently, for this case, the onset remained undefined. This illustrates the difficulty to defined accurate thresholds for complex fog life cycles.

L278: Finally, evolution of the optical depth revealed a strong increase at the transition during IOP 14, while lower values are found for IOPs 11 and 6. The threshold of 5 appears then too high, and we find that a value of 2 is more appropriate to discriminate between optically thin and thick fogs. Note that this value is consistent with retrievals reported by Guy

et al. (2023) from spectral measurements of downwelling longwave radiation performed with the Atmospheric Emitted Radiance Interferometer (AERI). For the 12 optically thin fogs sampled in central Greenland, including 9 mixed-phase cases, most of the 5-min liquid and ice optical depth retrievals are much lower than 2, with median values of 0.8 and 0.5, respectively.

*L470 (Section 4.2.1) Explain how the time interval between first and last threshold can be compared with the transition phase duration of Dione et al. (2023) for IOP14*

Using the methodology described in the response to Q2.2, the five transition endings are distributed between 2342 (CTH, orange) and 0030 UTC (TKE, green). The transition phase duration is then shorter than the period defined by Dione et al. (2023) that ends at 0100 UTC. However this ending value of 0100 UTC is not clear to us. Indeed as mentioned in their Fig. 10, the LWP and $\alpha^{closure}_{eq}$ are disrupted between 00:30 and 02:30 UTC, because of the stratus cloud above, CTH from the radar suddenly increased around 00:30, and the fog layer became unstable before midnight. Finally it is indicated in their Table 2 that the adiabatic phase started at 00:20, that is 40 minutes before the end of the transition phase, which seems not consistent.

We have nevertheless rephrased this sentence to emphasize on the transition phase duration :

L563: The transition phase duration is therefore short for this case, less than 1 h, and included in the period determined by Dione et al. (2023) which extends to 0100 UTC.

*L512 (Section 4.2.2) You write "Surprisingly, reverse LWC trends were observed for profiles # 6 and # 7 with α <0 even though the thin-to-thick transition had already occurred one hour earlier."*

*In fact, this is not surprising for a slow transition. The fact that thresholds are reached does not mean that transition is over and that fog has become adiabatic.*

*Discuss how the LWC profiles temporal evolution that you show in can explain the slow transition.*

As discussed before (Q2.1), despite all the thresholds are reached in a short period of time (25 minutes) for this case, the transition period is about 1h30-2h, and then effectively the transition is rather slow. But we think that the transition is over when all the thresholds are reached, that is around 0045 UTC, in contrast to Dione et al. 2023 that suggested that the transition ends later around 0230 UTC. Indeed, as discussed in the text, the profile #4 revealed that LWC is already increasing with height at 2330 UTC and the lower part of the profile #5 at 0045 UTC is very close to the adiabatic. The upper part of the profile #5 is difficult to explain (as stated in the discussion) but these two profiles suggest that the transition is over at this time. Then the reverse LWC trend found for profiles #6 and #7 are therefore surprising given that we expect more adiabatic features with LWC increasing with height. We then hypothesized that these two profiles are explained by sedimentation and collision-coalescence processes that redistribute LWC through the fog layer from the top to

the ground, as discussed in Sec. 5. As a result of these microphysical processes, the vertical development of fog is then delayed, and the various parameters, such as CTH, LWP, temperature stability and LW budget, remain constant until 02:30 UTC.

*L637, 638 "uncertainty periods" is used again for transition phase duration. Please revise.*

Done. We also have revised sentences L740-747 to discuss the differences in transition duration estimations between the IOPs 11, 14 and 6.

*Minor comments*

*Section 2.1. Check consistency in the tense used throughout this section (past, present, future).*
*Corrected.*

*L106 Change "spread" to "distributed"*
*Corrected*

L119: CBH cannot be derived unambiguously with a cloud radar alone as the signal in the lower part of the cloud can come from cloud droplets or from drizzle below the cloud. CBH is usually derived from a ceilometer.

We thank the reviewer for this comment.
The ceilometer is indeed more appropriate to derive cloud base height than the BASTA cloud radar due to uncertainties on the presence of drizzle. Sec. 2.1, Fig. 2 and Fig. 8c to 11c have been modified accordingly.

Information on CBH determination has been added as follows in Sec. 2.1 :

L131: Cloud base height (CBH) was determined using observations from a Vaisala CT25K 1/4 Hz ceilometer located at the Charbonnière site, measuring CBH for up to 3 cloud layers, with a vertical resolution of 15 m.

L129 "However, we analyse here all the data collected during the SOFOG3D experiment, and we then use independant retrieval." Not clear, please rephrase.
"independant" à "independent"
Corrected

The sentence has been rephrased as follows :

L138: Here we analyze independently the BASTA radar and HATPRO radiometer measurements collected during 12 IOPs of the SOFOG3D experiment.

L137 "aspirates" à "sucks in"
Corrected

Table 2. What objective criteria do you do define "radiative" vs "radiative-advective" fog types?

The classification is manual : "advective" characteristics were estimated for each case by a sudden increase in wind speed, longwave downward radiation and specific humidity of about 0.5 m.s$^{-1}$, 20 W.m$^{-2}$ and 0.3 g.kg$^{-1}$ respectively, as well as rapid variation in wind direction of more than 50°, and a sudden decrease in visibility of more than 15 km within a few minutes.

These details have been added to Sec. 2.2

L164: In particular, a sudden increase in longwave radiation, wind speed and specific humidity, as well as a decrease in visibility and variation in wind direction, reflecting advective processes, were analyzed for each case.

*L203 "These discrepancies may be explained by the contrasting environment between the two measurement areas (Thornton et al., 2023)." Explain what contrasts you are referring to ?*
Done. We have made the following changes :

L220: These discrepancies may be explained by the contrasting environment between the two measurement areas, characterized respectively by a complex topography for LANFEX with sites located in valleys and at hilltops, and a relatively flat area at SOFOG3D with a mixture of open and forested sites (Thornton et al., 2023).

L307 "These results highlight that while the adiabatic model correctly represents thermodynamical and microphysical properties of well-mixed fogs, it does not represent the properties of optically thin fogs at all." Does anyone expect the adiabatic model to correctly represent optically thin fog ?
The sentence has been modified as follows :

L364: These results highlight contrasting vertical profiles between well-mixed fogs, whose characteristics can be correctly represented by the adiabatic model, and thin fogs exhibiting an opposite trend.

L393 "This attests that differences observed from our dataset result from the actual properties of the fog sampled during SOFOG3D and not from the measurements used (in situ or remote sensing) to compute the fog adiabaticity from closure." Not clear, please rephrase or explain.
Done, the sentence has been rephrased as follows :

L467: This attests that lower values of $\alpha^{closure}_{eq}$ for well-mixed deep fogs in our dataset do not result from the type of instrument used to compute adiabaticity from closure (in situ or remote sensing) but reflect the actual properties of the fogs sampled during SOFOG3D.

L634 Replace "if" by "while".
Corrected

L703 "period of uncertainty" please rephrase according to Major comment 2 discussions.

Corrected, we have modified the sentence to emphasize on the uncertainty of the transition phase ending.

L799: We used remote sensing instruments (microwave radiometer and Doppler cloud radar) and surface measurements to determine the thin-to-thick transition ending time, based on 5 thresholds for LW flux, TKE, vertical temperature gradient, fog top height and LWP, enabling us to assess an associated interval of duration.

L703 "more suited …" than what ?
Corrected

L803: We found that a LWP threshold value of 15 $g.m^{-2}$ and an optical depth threshold value of 2, are more suited for the 4 thick fogs sampled at the Charbonnière site than the values of 30 $g.m^{-2}$ and 5, respectively, proposed by Wærsted et al. (2017).

L711 " larger negative values …" than what ?
Corrected, replaced by 'large'.

Finkelstein, P. L. and Sims, P. F.: Sampling error in eddy correlation flux measurements, Journal of Geophysical Research: Atmospheres,
106, 3503–3509, https://doi.org/https://doi.org/10.1029/2000JD900731, 2001

Guy, H., Brooks, I. M., Turner, D. D., Cox, C. J., Rowe, P. M., Shupe, M. D., Walden, V. P., and Neely III, R. R.: Observations of Fog-Aerosol Interactions Over Central Greenland, Journal of Geophysical Research: Atmospheres, 128, e2023JD038 718,
https://doi.org/10.1029/2023JD038718, e2023JD038718 2023JD038718, 2023.

Lenschow, D. H., Mann, J., and Kristensen, L.: How Long Is Long Enough When Measuring Fluxes and Other Turbulence Statistics?, Journal of Atmospheric and Oceanic Technology, 11, 661 – 673, https://doi.org/10.1175/1520-0426(1994)011<0661:HLILEW>2.0.CO;2, 1994

Wood, R.: Stratocumulus Clouds, Monthly Weather Review, 140, 2373 – 2423, https://doi.org/10.1175/MWR-D-11-00121.1, 2012.

---

## Author Response (AR3)

The authors would like to thank the editor for the time spent on the revision of the manuscript. As requested, we have revised the colour schemes with a colormap that allows readers with colour vision deficiencies to correctly interpret the findings, using the Coblis Color Blindness Simulator. Figures 8 to 12 and Fig. 14, which appear below have accordingly been modified with a uniform sequential colormap.

Editor report

**Please ensure that the colour schemes used in your maps and charts allow readers with colour vision deficiencies to correctly interpret your findings. Please check your figures using the Coblis – Color Blindness Simulator (https://www.color-blindness.com/coblis-color-blindness-simulator/) and revise the colour schemes accordingly.**

[Figure]

Figure 8. Temporal evolution during IOP 13b of (a) visibility, temperature at 2m and wind barbs at 10 m above ground level (empty circles, half barbs and full barbs represent wind speeds < 2.5, 5 and 10 m.s−1 , respectively. The orientation of the barbs gives the wind direction); (b) the upward (brown) and downward (red) longwave radiation. The gray shaded area delimitate foggy periods; (c) reflectivity and CTH derived from the BASTA cloud radar, and CBH (dashed line) from the ceilometer. The trajectory of the tethered balloon is superimposed in grey.
Each selected vertical profile is highlighted in blue and labeled by its number; (d) Corresponding vertical profiles of measured temperature, with adiabaticity (red) and adiabatic lapse rate (black); (e) Corresponding vertical profiles of LWC measured by the CDP with adiabaticity (red) and adiabatic (black); (f) Temporal evolution of the adiabaticity for the selected profiles : LWC (blue line) and lapse rate (green line).

[Figure]

Figure 9. Same as Fig. 8 but for IOP 14. In addition b) the onset time from Dione et al. (2023) and the different ending times of the thin-to-thick fog transition are indicated by vertical lines, as in Figure 2; c) and f) the onset time is indicated by a vertical grey dashed line and the ending time derived from longwave net radiation is indicated by a vertical red dashed line.

[Figure]

Figure 10. Same legend as Fig. 9 but for IOP 11

[Figure]

Figure 11. Same legend as Fig. 9 but for IOP 6. Visibility in (a) is issued from the Jachère site.

[Figure]

**(a)**

**N=140**

○ Thin Fog   ◆ Stratus
● Thick Fog

**(b)** LWP

**(c)** CTH

$LWP_{CDP}$ $(g.m^{-2})$

$CTH_{CDP}$ (m)

Figure 12. Adiabaticity parameter α as a function of γ in logarithmic scale, over the 12 IOPs : (a) colored with IOPs as in Fig. 4, thin and thick fog are indicated by empty dots and filled dots, respectively, (b) colored with LWP values issued from the CDP measurements, (c) colored with CTH values issued from the CDP measurements. Fog and stratus are indicated by dots and diamonds respectively.

**IOP 11**

**IOP 14**

$dN/dlogD$ $(cm^{-3})$

Diameter (μm)

— #4
— #5
— #6
— #7

— #3
— #4
— #5

Figure 14. Droplet number size distributions measured by CDP in the 15 m thick layer above the surface for profiles (a) # 4 to # 7 from IOP 11, and (b) # 3 to # 5 from IOP 14